



# XCO₂ retrieval for GOSAT and GOSAT-2 based on the FOCAL algorithm

Stefan Noël[1], Maximilian Reuter[1], Michael Buchwitz[1], Jakob Borchardt[1], Michael Hilker[1], Heinrich Bovensmann[1], John P. Burrows[1], Antonio Di Noia[2], Hiroshi Suto[3], Yukio Yoshida[4], Matthias Buschmann[1], Nicholas M. Deutscher[5], Dietrich G. Feist[6,7,8], David W. T. Griffith[5], Frank Hase[9], Rigel Kivi[10], Isamu Morino[4], Justus Notholt[1], Hirofumi Ohyama[4], Christof Petri[1], James R. Podolske[11], David F. Pollard[12], Mahesh Kumar Sha[13], Kei Shiomi[3], Ralf Sussmann[14], Yao Té[15], Voltaire A. Velazco[5], and Thorsten Warneke[1]

[1]Institute of Environmental Physics, University of Bremen, FB 1, P.O. Box 330440, 28334 Bremen, Germany
[2]Earth Observation Science, University of Leicester, LE1 7RH, Leicester, UK
[3]Japan Aerospace Exploration Agency (JAXA), 305-8505, Tsukuba, Japan
[4]National Institute for Environmental Studies (NIES), 305-8506, Tsukuba, Japan
[5]Centre for Atmospheric Chemistry, School of Earth, Atmospheric and Life Sciences, University of Wollongong NSW 2522 Australia
[6]Max Planck Institute for Biogeochemistry, Jena, Germany
[7]Deutsches Zentrum für Luft- und Raumfahrt, Institut für Physik der Atmosphäre, Oberpfaffenhofen, Germany
[8]Ludwig-Maximilians-Universität München, Lehrstuhl für Physik der Atmosphäre, Munich, Germany
[9]Karlsruhe Institute of Technology, IMK-ASF, Karlsruhe, Germany
[10]Finnish Meteorological Institute, Space and Earth Observation Centre, Tähteläntie 62, 99600 Sodankylä, Finland
[11]NASA Ames Research Center, Atmospheric Science Branch, Moffett Field, CA 94035, USA
[12]National Institute of Water and Atmospheric Research Ltd (NIWA), Lauder, New Zealand
[13]Royal Belgian Institute for Space Aeronomy (BIRA-IASB), Brussels, Belgium
[14]Karlsruhe Institute of Technology, IMK-IFU, Garmisch-Partenkirchen, Germany
[15]Laboratoire d'Etudes du Rayonnement et de la Matière en Astrophysique et Atmosphères (LERMA-IPSL), Sorbonne Université, CNRS, Observatoire de Paris, PSL Université, 75005 Paris, France

**Correspondence:** S. Noël (stefan.noel@iup.physik.uni-bremen.de)

**Abstract.**

Since 2009, the Greenhouse gases Observing SATellite (GOSAT) performs radiance measurements in the shortwave-infrared (SWIR) spectral region. From February 2019 onward, data from GOSAT-2 are also available.

We present first results from the application of the Fast atmOspheric traCe gAs retrieval (FOCAL) algorithm to derive
5   column-averaged dry-air mole fractions of carbon dioxide ($XCO_2$) from GOSAT and GOSAT-2 radiances and their validation. FOCAL has initially been developed for OCO-2 $XCO_2$ retrievals and allows simultaneous retrievals of several gases over both land and ocean. Because FOCAL is accurate and numerically very fast it is currently considered as a candidate algorithm for the forthcoming European anthropogenic $CO_2$ Monitoring (CO2M) mission, to be launched in 2025.

We present the adaptation of FOCAL to GOSAT and discuss the changes made and GOSAT specific additions. This includes
10   particularly modifications in pre-processing (e.g. cloud detection) and post-processing (bias correction and filtering).





A feature of the new application of FOCAL to GOSAT/GOSAT-2 is the independent use of both S and P polarisation spectra in the retrieval. This is not possible for OCO-2, which measures only one polarisation direction. Additionally, we make use of GOSAT's wider spectral coverage compared to OCO-2 and derive not only $XCO_2$, water vapour ($H_2O$) and solar induced fluorescence (SIF) but also methane ($XCH_4$), with the potential for further atmospheric constituents and parameters like semiheavy water vapour (HDO) and (in the case of GOSAT-2) also carbon monoxide (CO) total columns and possibly nitrous oxide ($XN_2O$).

Here, we concentrate on the new FOCAL $XCO_2$ data products. We describe the generation of the products as well as applied filtering and bias correction procedures. GOSAT-FOCAL $XCO_2$ data have been produced for the time interval 2009 to 2019. Comparisons with other independent GOSAT data sets reveal an agreement of long-term temporal variations within about 1 ppm over one decade; differences in seasonal variations of about 0.5 ppm are observed. Furthermore, we obtain a mean regional bias of the new GOSAT-FOCAL product to the ground based Total Carbon Column Observing Network (TCCON) of 0.56 ppm with a mean scatter of 1.89 ppm.

The GOSAT-2-FOCAL $XCO_2$ product is generated in a similar way as the GOSAT-FOCAL product, but with adapted settings. All GOSAT-2 data until end of 2019 have been processed. Because of this limited time interval, the GOSAT-2 results are considered to be preliminary only, but first comparisons show that these data compare well with the GOSAT-FOCAL results.

## 1 Introduction

Carbon dioxide ($CO_2$) is the most important greenhouse gas in the context of global warming (e.g. IPCC, 2013). The amount of $CO_2$ in the atmosphere is primarily determined by natural and anthropogenic sources and sinks but our current understanding of these sources and sinks has significant gaps (e.g., Ciais et al., 2014; Reuter et al., 2017a; Friedlingstein et al., 2019; Janssens-Maenhout et al., 2020). Retrievals of column averaged carbon dioxide ($XCO_2$) from the satellite sensors SCIA-MACHY/ENVISAT (Burrows et al., 1995; Bovensmann et al., 1999; Reuter et al., 2010, 2011), TANSO-FTS/GOSAT (Kuze et al., 2016) and from the Orbiting Carbon Observatory-2 (OCO-2) satellite (Crisp et al., 2004; Eldering et al., 2017; O'Dell et al., 2012, 2018) have been used in over a decade to obtain information on natural $CO_2$ sources and sinks (e.g., Chevallier et al., 2014; Chevallier, 2015; Reuter et al., 2014b, 2017a; Schneising et al., 2014; Basu et al., 2013; Houweling et al., 2015; Kaminski et al., 2017; Liu et al., 2017; Eldering et al., 2017; Yin et al., 2018; Palmer et al., 2019) and on anthropogenic $CO_2$ emissions (e.g., Schneising et al., 2008, 2013; Reuter et al., 2014a, 2019; Nassar et al., 2017; Schwandner et al., 2017; Miller et al., 2019; Labzovskii et al., 2019; Wu et al., 2020; Zheng et al., 2020)

First satellite measurements of $XCO_2$ were performed by the Scanning Imaging Absorption Spectrometer for Atmospheric CHartographY (SCIAMACHY) instrument (Bovensmann et al., 1999; Gottwald and Bovensmann, 2011; Reuter et al., 2010, 2011) on the European environmental satellite ENVISAT launched in 2002 and operating until April 2012.

Whereas greenhouse gases were only one field of application among others of SCIAMACHY, later satellite missions focused explicitly on these. In 2009, the Greenhouse gases Observing SATellite (GOSAT; Kuze et al., 2009, 2016) was launched,





followed by the Orbiting Carbon Observatory-2 (OCO-2; Crisp et al., 2017; Eldering et al., 2017; O'Dell et al., 2012, 2018) in

2014. Furthermore, in 2016 the Chinese TanSat mission was launched; first results have been presented by Yang et al. (2018). Follow-on instruments to GOSAT and OCO-2 (GOSAT-2; Suto et al., 2020) and (OCO-3; Eldering et al., 2019) are in orbit since 2018 and 2019, respectively. TanSat, GOSAT and OCO-2/3 instruments are still operating, and several different retrieval algorithms have been developed to derive $XCO_2$ from their short-wave infrared (SWIR) spectra.

The main challenge for space-borne $XCO_2$ measurements is the required accuracy of the resulting data products as the

atmospheric background of $XCO_2$ is high compared to the variability, which is typically less than a few percent (about 2% seasonal cycle variations in the northern hemisphere in addition to an annual increase of about 0.5% per year, see e.g. Schneising et al., 2014; Buchwitz et al., 2018). Depending on the application, even higher accuracies are needed. An accurate $XCO_2$ retrieval usually requires a complex retrieval method and large computational effort. This is no major problem for the number of measurements provided by the GOSAT instruments, but even current OCO-2 retrievals require significantly larger compu-

tational effort. However, new missions with much higher spatial resolution and coverage are currently in preparation to answer the challenging questions on $CO_2$ local and global sources and sinks in a changing climate, one amongst them is the forth-coming European anthropogenic $CO_2$ Monitoring (CO2M) mission (Kuhlmann et al., 2019; Janssens-Maenhout et al., 2020), dramatically increasing the computational power needed for retrievals.

Three years ago, Reuter et al. (2017b, c) developed the Fast atmOspheric traCe gAs retrieval (FOCAL) and applied it to

OCO-2 data. To show the applicability of the FOCAL method not only to OCO-2 but also to other satellite sensors, we present in this study a new application of FOCAL to GOSAT and also some first results from an application to GOSAT-2. GOSAT-FOCAL has several advantages over GOSAT-BESD (Heymann et al., 2015), the currently used IUP GOSAT XCO2 retrieval, product (Heymann et al., 2015), which provides only $XCO_2$ data over land. However, FOCAL is able to retrieve not only $XCO_2$ but – depending on the used spectral ranges – also other atmospheric parameters like $XCH_4$, $H_2O$, HDO, CO and

$N_2O$. In the present study we concentrate on $XCO_2$, as this is the most important (and because of its high requirements on accuracy possibly most challenging) anthropogenic greenhouse gas.

The manuscript is organised as follows: In Section 2 we list all data sets used in this study. The retrieval algorithm is described in section 3. Sections 4 and 5 then show the results of the retrieval and the validation. Finally, the conclusions are given in section 6.

# 2   Data sets used

## 2.1   GOSAT and GOSAT-2

The Greenhouse gases Observing SATellite (GOSAT; Kuze et al., 2009) was launched in January 2009 and is still in operation. The Thermal And Near infrared Sensor for carbon Observation (TANSO) on-board GOSAT consists of a Cloud and Aerosol Imager (TANSO-CAI) and a Fourier Transform Spectrometer (TANSO-FTS), which measures radiances in the SWIR spectral

region with S and P polarisation and in the thermal infrared spectral region without polarisation with a spectral resolution of





$0.2\,cm^{-1}$. The FOCAL retrieval uses as main input calibrated GOSAT L1B V220.220 spectra from the three SWIR bands (around 0.76, 1.6 and 2.0 μm) of TANSO FTS.

GOSAT-2 (Nakajima et al., 2017; Suto et al., 2020) was launched in October 2018 and comprises a similar instrumentation as GOSAT. The GOSAT-2 FTS has the same spectral resolution but an extended spectral range for SIF and CO retrievals.

We use calibrated GOSAT-2 L1B SWIR data V101.101.

Both GOSAT and GOSAT-2 perform point measurements with a spatial resolution (footprint diameter) of about 10 km. For both instruments, we use a tabulated instrumental line shape (ILS) with a kernel width of $15\,cm^{-1}$. For GOSAT this has been generated by a theoretical formula parameterising a "real-world" FTS instrument (see e.g. formula 5.21 in Davis et al., 2001) , which depends on the maximum optical path difference (MOPD, ±2.5 cm for GOSAT) and the size of the instantaneous field

of view (IFOV, 15.8 mrad for GOSAT). The same formula has been used by Heymann et al. (2015). This ILS is symmetric and the same for S and P polarisation.

For GOSAT-2, we use a preliminary tabulated ILS provided by JAXA and generated on 16 January 2020, which is different for S and P polarisation and asymmetric, especially in the SWIR-1 band. Meanwhile, this ILS has been officially released and is available via the NIES web site.

## 2.2 Reference Spectra and External Databases

For the retrieval several reference spectra and databases are used.

The solar spectrum used in the forward model is based on a high resolution solar transmittance spectrum (O'Dell et al., 2012) in combination with an ISS solar reference spectrum (Meftah et al., 2018). For the SIF retrieval we used a chlorophyll fluorescence spectrum by Rascher et al. (2009), which has been scaled to $1.0\,mW/m^2/sr/nm$ at 760 nm.

We use tabulated cross sections at a $0.001\,cm^{-1}$ sampling based on HITRAN2016 (Gordon et al., 2017) and the absorption cross section database ABSCO v5.0 (Benner et al., 2016; Devi et al., 2016) from the NASA (National Aeronautics and Space Administration) ACOS/OCO-2 project.

Surface elevation, surface roughness and surface type are derived from the Global Multi-resolution Terrain Elevation Data (GMTED2010; Danielson and Gesch, 2011) of the U.S. Geological Survey (USGS) and the National Geospatial-Intelligence

Agency (NGA) at a spatial resolution of 0.025°. Meteorological information (pressure, temperature, water vapour profiles) is obtained from ECMWF (European Centre for Medium-range Weather Forecasts) ERA 5 model data (Hersbach et al., 2020), which are available every 1 hour on a 0.25° horizontal grid and on 137 altitude layers.

We use $XCO_2$ data from the CarbonTracker (CT) model CT2019 and CT-NRT v2020-1 (Jacobson et al., 2020a, b) and data from the Total Carbon Column Observing Network (TCCON, see e.g. Wunch et al., 2011) in the context of the bias correction

database (see section 2.3). TCCON data are also used for validation (see section 5). Table 1 lists the TCCON stations which provided data for the present study.

$XCO_2$ a-priori profiles are derived using the 2018 version of the simple empirical $CO_2$ model SECM (Reuter et al., 2012). In the context of validation, we use the 2020 version of SECM. $XCH_4$ a-priori data are from the simple $CH_4$ climatological model SC4C2018 developed and used by Schneising et al. (2019) and briefly described by Reuter et al. (2020).





For $CO_2$ we use the "synth" a-priori error covariance matrix described by Reuter et al. (2017b). For $H_2O$, we use the same error covariance matrix as Reuter et al. (2017b), but scaled by a factor of 5 to reduce the dependencies of the retrieval results on the a-priori. For $CH_4$, for convenience, we scale the $CO_2$ matrix to result in an $XCH_4$ uncertainty of 45 ppb, which is considered to be a reasonable estimate. Note that only the matrices are scaled, not the a-priori values.

## 2.3    The "true" database

Quality filtering and bias correction usually require the knowledge of a "true" (in this case $XCO_2$) value. For this, we do not simply use model data as truth, as one aim of $XCO_2$ products is to improve models. Another method is to take ground-based TCCON measurements as basis for a bias correction. However, although TCCON measurements are very accurate, they are only available at certain locations and are therefore more suited for validation.

    Our choice is therefore to use a data base generated from a combination of TCCON measurements and CarbonTracker (CT)
model data for a reference year (2018 for GOSAT, 2019 for GOSAT-2).

    This database is produced in the following way: As a first step, we determine from the CT data global daily 3D maps close to 13:00 local time (i.e. GOSAT and GOSAT-2 equator crossing time). We reduce the altitude grid to five layers with the same dry-air sub-columns, i.e. the same amount of particles, and interpolate the data from the native CT horizontal resolution of $3° \times 2°$ to $0.5° \times 0.5°$. Then we determine from the TCCON data daily mean values ($XCO_2{}^{TCCON}$) for $13\,h \pm 2\,h$ local time. Next,
we select collocated CT data and correct them for the TCCON averaging kernels, resulting in a TCCON corrected CT value at the TCCON location ($XCO_2{}^{CT}$). The application of the averaging kernels corrects for different vertical resolutions/sensitivities (see e.g. Rodgers and Connor, 2003; Wunch et al., 2010). We look for contiguous regions where CT data differ by less than $0.75\,ppm$ from $XCO_2{}^{CT}$; these data are then used for the "true" database. The result are daily maps containing $CO_2$ data for five vertical sub-layer altitudes. The spatial coverage is usually not global and varies from day to day. There are typically more
data in the southern hemisphere during the second half of the year. When comparing with GOSAT or GOSAT-2 measurement results, the "true" $XCO_2$ is then computed from the $CO_2$ layers of the true database, considering the retrieval's averaging kernels.

    Specifically, we use here for GOSAT CT2019 data in combination with TCCON GGG2014 (see Tab. 1) for 2018. For GOSAT-2 we also use TCCON GGG2014 data, but need to rely on CT-NRT v2020-1 for 2019. Because the CT NRT data are
not yet available for the whole year 2019, the GOSAT-2 "true" database does not cover the whole year; there are essentially no data after August 2019. This is a limiting factor for GOSAT-2, especially because this also means that data in the southern hemisphere are less present in the 2019 database.

    Please note that the "true" database does not contain any TCCON data - it only contains CT data which were confirmed by TCCON, but individual values may differ by up to $1.5\,ppm$. This is why a later validation with TCCON still makes sense.

## 2.4    GOSAT Level 2 Products

To assess the quality of the newly created GOSAT-FOCAL $XCO_2$ products, they have been compared with several other well-established GOSAT Level 2 data sets (see section 5). The GOSAT BESD v01.04 product from IUP (Heymann et al., 2015) is a





near-real time product generated in the context of the Copernicus Atmospheric Monitoring Service (CAMS, https://atmosphere.copernicus.eu/ (last access: 30-July-2020)) project. It is available from 2014 onward. The GOSAT RemoTeC v2.3.8 product

from SRON (Butz et al., 2011) and the full-physics GOSAT product from the University of Leicester v7.3 (Cogan et al., 2012) were generated in the context of the Copernicus Climate Change Service (C3S, https://climate.copernicus.eu/; last access: 30-July-2020) and cover the GOSAT time series from 2009 until end of 2019. The recently released NASA GOSAT ACOS v9r product (O'Dell et al., 2012, 2018; Kiel et al., 2019) is also available for the years 2009 to 2019. The operational GOSAT $XCO_2$ product v02.95 (bias corrected) from NIES currently ends in August 2020. The BESD product contains only $XCO_2$

data over land, all other products are available for water and land surfaces.

## 3   Retrieval Algorithm

The retrieval is performed in three main steps: Pre-processing, processing and post-processing. These are described in the following sub-sections.

### 3.1   Pre-Processing

During pre-processing all required input data for the main processing step are collected. Furthermore, a first filtering of data is performed to reduce processing time.

The pre-processing procedure is largely based on the pre-processing as present in the BESD GOSAT product (Heymann et al., 2015). The sequence of pre-processing activities is as follows:

1. Extraction of measured spectra, geolocation and information on quality and measurement modes (e.g. gain, scan direc-
tion) from the GOSAT L1B product.

2. Estimation of instrument noise and cloud parameters.

3. Filtering for data quality, latitudes, solar zenith angle, signal-to-noise ratio and clouds (see Tab. 2 for settings).

4. Extraction of surface type, elevation and roughness derived from the surface database for each measurement.

5. Addition of corresponding meteorological information (pressure, temperature, dry-air column and water vapour profiles)
for the time and place of the measurements. This includes a correction for surface elevation, i.e. model profiles are extended / cut according to the value from the surface database.

6. Add a-priori gas profiles for each measurement ($CO_2$ from SECM, $CH_4$ from SC4C, $H_2O$ from meteorology). For GOSAT-2, also a-priori profiles for CO and $N_2O$ are added. The latter do not depend on geolocation; they are based on the tropical reference atmosphere from Anderson et al. (1986), scaled to column average values of XCO = 0.1 ppm and
$XN_2O = 330$ ppb.





Because FOCAL is a fast algorithm and the number of GOSAT and GOSAT-2 measurements is much less than for OCO-2, we chose to set the pre-processing filters relatively relaxed and to apply the quality filtering mostly in the post-processing. As can be seen from Fig. 1 about two thirds of the measurements are filtered out during pre-processing.

### 3.1.1 Noise Estimate

Similar to Heymann et al. (2015) the spectral noise is initially assumed to be independent from wavenumber for each band. It is estimated from the standard deviation of the real part of the "dark" off-band signal (i.e. the first 500 spectral points in each band). In a later step (see Section 3.2.1) this noise will be modified to account for additional forward model errors and overall scaling.

### 3.1.2 Cloud Filter

The cloud filtering is based on two physical properties of clouds: clouds are (usually) bright and clouds are high (higher than the surface) so that little water vapour is above them. In the pre-processing these properties are described by two quantities: cloud albedo and water vapour path. These are derived for each spectrum as described in Heymann et al. (2015). The cloud albedo for each band is estimated from the mean reflectance $L$ within a spectral range outside the absorption. $L$ is determined from the mean radiance $I$, the mean irradiance $I_0$ and the solar zenith angle $\alpha$ via:

$$L = \frac{\pi I}{I_0 \cos \alpha} \tag{1}$$

The specific wavenumber ranges and irradiance values used for filtering are given in Tab. 3.

The water vapour path is determined from a spectral region with strong water vapour absorption in the SWIR-3 band (see Tab. 3). It is given by the ratio between the median radiance and the median of the estimated noise in this spectral range.

A ground pixel is assumed to be cloudy if either the cloud albedo in one of the bands or the water vapour path exceeds the thresholds given in Tab. 2.

### 3.2 Processing

The processing is based on the Fast atmOspheric traCe gAs retrievaL (FOCAL) algorithm which is described in detail in Reuter et al. (2017c). A first successful application of this algorithm to OCO-2 data is given in Reuter et al. (2017b). Therefore, we only summarise the main features of the algorithm here and point out the differences to the OCO-2 application.

FOCAL appoximates modifications of the direct light path due to scattering in the atmosphere by a single scattering layer, which is characterised by its height (pressure level), its optical thickness and an Ångström parameter which describes the wavenumber dependence of scattering. The layer height is normalised to the surface pressure. Furthermore, Lambertian scattering on the surface is considered. For atmospheric scattering processes an isotropic phase function is assumed. With this approximation, the FOCAL forward model is essentially an analytical formula; it uses pre-calculated and tabulated cross sections such that calculations can be performed considerably fast. The inversion of the forward model is based on optimal estimation (Rodgers, 2000) and uses the Levenberg-Marquardt-Fletcher method (Fletcher, 1971) to minimise the cost function.





The OCO-2 retrieval of Reuter et al. (2017b, c) uses four fit windows in the NIR (near-infrared) and SWIR spectral range to derive the atmospheric parameters $XCO_2$, water vapour and SIF. In contrast to OCO-2, GOSAT and GOSAT-2 cover a wider spectral range and provide spectra in two polarisation directions referred to as S and P. Therefore, we treat in our retrieval

both polarisation directions as independent spectra opposed to the average of both as usually used in other GOSAT retrievals (see e.g. Butz et al., 2011; Cogan et al., 2012; O'Dell et al., 2012). However, recently Kuze et al. (2020) presented a methane retrieval for GOSAT based on an algorithm from Kikuchi et al. (2016), which also makes use of both polarisation directions. Furthermore, the FOCAL fitting windows (see Tab. 4) have been adapted to the specific GOSAT(-2) spectral bands such that in addition also other atmospheric constituents and parameters like HDO and (in the case of GOSAT-2) also CO total columns

and possibly $XN_2O$ can be retrieved. This results in six fitting windows for GOSAT and eight windows for GOSAT-2 for each polarisation. The retrieval is performed on a wavenumber axis.

Because of the large number of target gases and spectral bands the retrieval requires various state vector elements. These are listed together with the fit windows, from which they are determined, and their a-priori values and uncertainty ranges in Tab. 5 for GOSAT and GOSAT-2.

For GOSAT, the retrieval determines $CO_2$, $CH_4$ and $H_2O$ on 5 layers with same number of air particles, from which then the column average values $XCO_2$, $XCH_4$ and $XH_2O$ are calculated. Furthermore, solar induced fluorescence (SIF) is determined by scaling of a corresponding reference spectrum.

Instead of the HDO column, we fit a scaling factor for the relative abundance of HDO compared to $H_2O$, $\delta D$, which is defined as:

$$\delta D = \frac{R_{\mathrm{meas}}}{R_{\mathrm{VSMOW}}} - 1 \tag{2}$$

where $R_{\mathrm{meas}}$ is the ratio of the measured HDO and $H_2O$ columns, $R_{\mathrm{VSMOW}}$ ( $= 3.1152 \times 10^{-4}$) is the corresponding value for Vienna Standard Mean Ocean Water (VSMOW). $\delta D$ is usually given in units of per-mill.

$\delta D = 0‰$ corresponds to HDO concentrations as in VSMOW, $\delta D = -1000‰$ to no HDO. We assume the same profile shape for HDO as for $H_2O$. For GOSAT-2, we also fit scaling factors to (fixed) CO and $N_2O$ profiles.

As mentioned above, atmospheric scattering is considered in FOCAL by a single scattering layer, which is described by three parameters (height, optical depth and Ångström coefficient). As scattering is different for S and P polarised light, we fit two independent layers for S and P.

In addition, we determine in each fit window (independently for S and P) a polynomial background function describing the surface albedo. For this we use second order polynomials except for the small SIF windows (no. 1) where a linear function is

230 sufficient.

The GOSAT data files only contain a fixed spectral axis. As e.g. described in Heymann et al. (2015), the spectral calibration of GOSAT changes especially at the begin of the mission with time. This change can be corrected by a spectral scaling factor. We determine this overall scaling factor by a spectral fit in the SIF window before the retrieval. So far, this spectral pre-fitting seems to be unnecessary for GOSAT-2. In the retrieval, we then additionally consider for both GOSAT and GOSAT-2 possible





additional spectral shifts and squeezes in each fit window by corresponding state vector elements, but the influences of these
spectral changes on the results is rather small.

### 3.2.1  Noise Model

The noise $N$ derived from the off-band signal is only an estimate. It does not consider a possible wavenumber dependence of
the noise within one spectral band. Furthermore, a potential error of the forward model needs to be considered. In the optimal
estimation method this can be achieved by including the forward model error in the measurement error covariance. For this,
we define a scaling factor $s$ for the estimated noise and the quantity $\delta F$, which denotes the relative error of the forward model.
The forward model error is proportional to the continuum radiance outside the absorption $I$, which is estimated from the 0.99
percentile of the measured radiance at the edge of each fit window. The quantities $\delta F$ and $s$ are determined using the approach
described in (Reuter et al., 2017b), i.e. by running the retrieval for a representative subset of data and then fitting the function

$$RSR(NSR) = \sqrt{(s\,NSR)^2 + \delta F^2} \tag{3}$$

to binned values of the residual-to-signal ratio (RSR) as function of the noise-to-signal ratio (NSR). RSR is defined as the
standard deviation of the retrieved spectral residual in each fit window divided by the continuum signal $I$; NSR is the standard
deviation of the noise divided by $I$.

With the method described in Reuter et al. (2017c) it is also possible to define a $2\sigma$-outlier limit based on NSR and RSR
data, which will be used to filter out too noisy data during post-processing (see section 3.3). This is parameterised by a second
order polynomial as a function of the uncorrected NSR

$$f_N(NSR) = a_0 + a_1\,NSR + a_2\,NSR^2 \tag{4}$$

which is added to the RSR function of Eq. (3). The coefficients $a_i$ are determined via a fit. To avoid extrapolation, $f_N$ is set to
the edge values outside the fitting range.

In order to cover the varying signal over the year, we base the noise model fits on data from one day per month for one
reference year. For GOSAT, we take from December 2017 to November 2018 (as there are only few GOSAT data available in
December 2018). For GOSAT-2 we use data from February 2019 to December 2019. In the case of GOSAT-2 we further restrict
the input data for the noise model parameter fit to data over land because some of the data over water show an unexpected
behaviour (low RSR in case of large NSR), which needs further investigation. In this sense, the current GOSAT-2 noise model
is considered to be preliminary and may need some refinement in the future.

Figs. 3 to 6 show the noise model results for GOSAT and GOSAT-2. The orange line gives the fitted RSR function, the
red line the outlier limit. The derived values from the noise model are given in Tab. 6 and 7 for GOSAT and GOSAT-2. The
forward model errors $\delta F$ are on average slightly larger for GOSAT-2 than for GOSAT. In the SWIR, values similar to OCO-2
are obtained, but in the NIR the OCO-2 $\delta F$ is typically smaller (about 0.003). This indicates that for GOSAT and GOSAT-2
instrumental/calibration effects seem to impact the radiance errors more in the NIR than in the SWIR.





### 3.3 Post-Processing

The purpose of post-processing is to filter out invalid data and to perform a bias correction for the products. The current post-processing focuses on $XCO_2$. The post-processing is performed in several steps, namely:

1. Basic filtering based on physical knowledge.

2. Filtering out low quality data using parameters / limits determined using a random forest classifier.

3. Application of a bias correction using a random forest regressor.

4. Additional filtering out of data with too large bias correction.

These steps are described in the following subsections.

### 3.3.1 Basic filter

The basic filtering removes measurements where the retrieval does not converge or where the quality of the fit results is too low. We consider this to be the case if the $\chi^2$ calculated over all fit windows is larger than 2 or if for at least one of the fit windows the RSR outlier limits (see section 3.2.1) are exceeded. Furthermore, we apply some initial filters for nonphysical values on the derived scattering parameters (i.e. layer height outside the atmosphere, Ångström coefficient not within [1,5]. We also limit the maximum allowed optical depth of the scattering layer to 0.02 to filter out too thick clouds or aerosol amounts
and use a maximum allowed $XCO_2$ error of $2\,\mathrm{ppm}$. As described by Reuter et al. (2017c), FOCAL simulates scattering only for an isotropic phase function. The prominent forward peak, usually existing for Mie scattering phase functions of cloud and aerosol particles does basically not modify the lightpath. As FOCAL's optical depths of the scattering layer do not include this forward peak, these optical depths are much smaller than optical depths including a strong forward peak while having a similar influence on the light path modification (see discussion in the publication of Reuter et al. (2017c)). The maximum value of
0.02 for the layer optical depth should therefore not be interpreted as e.g. an aerosol optical depth.

The limits for the optical depth of the scattering layer and the $XCO_2$ error are somewhat arbitrary and actually result from visual inspection of the retrieval results. However, they are only intended as a first rough quality filter to facilitate later filter and bias correction methods, which will partly use the same parameters (see below). The detailed choice of these limits is therefore considered uncritical for the final results.

The above mentioned filter parameters and limits (see Tab. 8) are applied to both land and water surfaces and are the same for GOSAT and GOSAT-2, except for the RSR outlier limit which has been determined individually for each instrument. Figs. 1 and 2 show exemplary how many data points are filtered out in this step.

### 3.3.2 Random forest filter

In the next step, data are filtered out based on their expected $XCO_2$ bias i.e. the difference to a "true" $XCO_2$. Of course,
this true $XCO_2$ value is normally not known. We therefore use the "true" reference database (as described in section 2.3)



to train a random forest classifier (Pedregosa et al., 2011) to identify those variables which would remove – in combination with a corresponding random forest database – a pre-described percentage $p$ of data based on their $XCO_2$ bias. This is done independently for data over land and water. Note that we are only interested here in the $XCO_2$ bias on top of an overall global bias as the latter will be handled via the bias correction.

We determine the list of relevant variables and the random forest database for the filtering in the following way: We use the (uncorrected) results of the retrieval for the reference and apply the basic filtering as described in section 3.3.1. Then, the subset of these data is selected which has a corresponding "true" value in the reference database. For these data we determine the $XCO_2$ bias (measurement - reference $XCO_2$) and subtract the monthly global median of this bias. We then sort the data according to this bias and flag those $p$ percent of data with the highest absolute bias values as "bad". The random forest classifier

is then trained by using randomly 90% of these data as input. The training is done in two iterations: First, with a complete set of possible input variables ("features") and output variables ("estimators"); then, using only a reduced set consisting of the 10 best features/estimators (i.e. those with highest random forest score of the first run). The random forest classifier then decides for each measurement based on these 10 variables if it is filtered out or not.

The initial list of possible features/estimators includes essentially all quantities available after the retrieval, including viewing

angles, surface properties and continuum signal for each fit window. Furthermore, the retrieved values of the state vector elements and their errors are included in this list as well as averaging kernels for the profiles. We explicitly exclude the geolocation of the measurement (latitude, longitude) and the retrieved values (but not the errors) for the data products we are interested in, i.e. the gases and SIF. This is to avoid e.g. the filtering out of certain geographical regions or removing all points with high $XCO_2$ values. However, we include as possible filter variable the gradient of the retrieved $CO_2$ profile (i.e. the

difference between the two lowermost layers) as this has shown to be a suitable quantity.

The original number of candidate variables presented to the random forest classifier is quite high (193 for GOSAT and 246 for GOSAT-2) as can be seen from Figs. 7 and 8 (top left plots), but there are only few with a high relevance. The ten best variables selected partly differ for land and water surface (as shown in the middle and left top panels), but they usually comprise scattering parameters, polynomial coefficients, spectral corrections and some $XCO_2$ related parameters.

The other 10% of the input data are used to test the performance of the classifier. The results from this test and other cross-validation activities indicate, that the random forest classification is – depending on surface – only accurate in about two thirds of the cases. This means that the filtering also removes possibly valid data points and does not remove all possibly bad ones. However, we do not expect a perfect classification, because it is not possible to describe all inter-dependencies via the set of input features.

To obtain a high quality of the remaining $XCO_2$ data, we therefore need to filter out quite a large percentage of data (and perform an additional filtering at a later time, see below). For future data products further investigations are planned to improve the performance of the classifier, e.g. by providing additional features from combination of existing ones (like the already used $CO_2$ gradient). The percentage $p$ of data to be filtered out is usually a trade-off between data quality and remaining amount of data. In the present case a 50% limit has been selected. Actually, as can be seen from Figs. 1 and 2, the relative amount of data

filtered out via the random forest classifier is not exactly 50% of the data remaining after the previous filters.

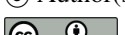



### 3.3.3 Bias correction and filtering

Reuter et al. (2017c) use for OCO-2 a bias correction based on the "small area approximation"(see also O'Dell et al., 2018; Kiel et al., 2019), which assumes that the variation of $XCO_2$ within a small area is small. This is not possible for GOSAT and GOSAT-2 because of their sparse sampling. We therefore follow a different approach here.

335 For the bias correction we use as input the same data set as for the random forest filter, but with this filter applied. 50% of the resulting data set is then used to train a random forest regressor, which aims to minimise the "true" $XCO_2$ bias (without global median subtracted) as function of the specified features. To create the bias correction database and the corresponding list of best features we again run the training twice, first with the full list (the same as for the filter) and then with the top ten features. Again, we use different corrections for land and water. The resulting parameters and their performance are shown

340 in the bottom panels of Figs. 7 and 8. The bias correction selects similar best features as the filter, but not exactly the same quantities in the same sequence.

 During application of the bias correction, the random forest regressor estimates the $XCO_2$ bias based on the values of the input variables. This bias is then subtracted from the retrieved value. Application of the bias correction to the training data set and the other 50% of the input data shows a comparable reduction of the $XCO_2$ scatter, which is an indication for a good

345 performance (e.g. no over-fitting) of the regressor.

 Currently, there is only a bias correction for $XCO_2$, but in principle this method is applicable also to other quantities depending on the availability of a corresponding "true" database.

 After the bias correction there are still a few outliers left in the $XCO_2$ data. These are filtered out by an additional filter on the derived $XCO_2$ bias. The limits for this filter are the global median bias for test data set $\pm 2\,\mathrm{ppm}$. the median bias is

350 different for land and water surfaces and also for GOSAT and GOSAT-2. The actual limits are given in Tab. 9. The value $2\,\mathrm{ppm}$ is estimated from visual inspection of the data. Figs. 1 and 2 show that typically less than 1–2% of the remaining data (less than 0.1% of all) are affected by this last filter.

## 4 Results

The FOCAL retrieval has been applied to all GOSAT and GOSAT-2 measurements until end of 2019. On average, FOCAL

355 needs $22\,\mathrm{s}$ with 6 iterations for the processing of one GOSAT ground pixel. For GOSAT-2 numbers are slightly larger ($28\,\mathrm{s}$/7 iterations) because of the additional fit windows and state vector elements. All performance values are given for a single core of an Intel Xeon E5-2667v3 CPU ($3.2\,\mathrm{GHz}$). These numbers are actually about one magnitude larger than the ones given in Reuter et al. (2017b, c) for the FOCAL application to OCO-2. This is because we use for GOSAT(-2) separate S and P polarisation spectra and more retrieved variables, which requires more and larger fit windows. For each of these fit windows

360 and parameters, weighting functions have to be calculated, which involves a convolution with the ILS. This convolution is the most time consuming part of the FOCAL retrieval. This is even more relevant for GOSAT(-2), because the FTS ILS is in principle sinc-shaped, i.e. it has a sharp peak in the centre but wide wings, which requires a large kernel width (in our case $15\,\mathrm{cm}^{-1}$ for the convolution.



Figs. 9 to 12 show examples for measured and fitted nadir mode radiance spectra for GOSAT and GOSAT-2 over land in the
365 different fitting windows. Since the difference between measured and modelled spectra is small and thus hard to see, we show
in Figs. 13 to 16 the corresponding residuals and the estimated noise. The residuals are on the order of magnitude of the noise,
which is slightly higher for P polarisation than for S polarisation. Some small spectral structures are visible in the residuals,
they appear more clearly in the smoothed residuals (convoluted with a 21 pixel boxcar), e.g. for GOSAT and GOSAT-2 in
the $O_2(A)$ band (window 2), and some broadband oscillations in window 4 and 5 for GOSAT-2. These features are present in
both S and P polarisations and occur also in other measurements, so they seem systematic. A reduction of these features could
possibly further improve future products.

In Fig. 17 some statistical information about the GOSAT-FOCAL data products is given. A time series for the number of
valid data is given in the top plot. In the recent years, about 5–6% of the available measurements could be transferred to
valid $XCO_2$ data. The number of valid data points increases from 2009 to 2019. This is mainly due to an increase in the data
over water, which is most likely related to optimisations in GOSAT operations (better use of glint geometry) over water. As
expected, the mean global $XCO_2$ shown in the middle plot increases with time. Global mean values over water are typically
slightly higher than over land; this is most likely a spatial sampling issue. The observed $XCO_2$ variability (standard deviation,
bottom plot) is larger over land which is attributed to influences of surface elevation. For GOSAT-2, only retrieved data from
2019 are available so far. The total amount of available measurements is about 2.8 million, compared to about 3.5 million
GOSAT measurements in 2019. Only about 3% of the GOSAT-2 data remain after all filtering / post-processing, which is
roughly half of the corresponding number for GOSAT (but similar to the first year of GOSAT). As can be seen from Fig. 2
more GOSAT-2 data are filtered out due to failed or bad convergence and by the RSR outlier limits than for GOSAT (Fig. 1).
Future improvements of the GOSAT-2 calibration or the noise model could possibly help here.

For further analyses, we have generated monthly maps on a $5° \times 5°$ grid. Example plots for the months April and August
2019 (begin/end of the growing season) are shown in Fig. 18 for GOSAT and in Fig. 19 for GOSAT-2. The data are not filtered
for low amounts of input data in the grid points, which may explain some individual outliers in the plots. Overall, the spatial
patterns observed by GOSAT and GOSAT-2 look reasonable. The north–south gradient in $XCO_2$ is visible with different sign
in April and August for both instruments. The spatial coverage of the GOSAT-2 data is lower than for GOSAT, because more
data are filtered out (see above). This seems to affect especially regions like the northern part of Africa.

**5   Verification and Validation**

For the verification and validation of the GOSAT and GOSAT-2 FOCAL products we perform a comparison with various
reference data sets (see section 2), namely:

–  The GOSAT BESD v01.04 product from IUP (referred to as "BESD" later).

–  The GOSAT ACOS v9r product from NASA (referred to as "ACOS" later).

–  The GOSAT UoL_FP v7.3 product from the University of Leicester (referred to as "UoL" later).





- The GOSAT RemoTeC v2.3.8 product from SRON (referred to as "SRON" later).

- The GOSAT operational product v02.95 (bias corrected) from NIES (referred to as "NIES" later).

- Collocated TCCON GGG2014 data (referred to as "TCCON" later).

For the comparisons, all data have been adjusted using the same a-priori (SECM2020).

Since all GOSAT products use different retrieval and filter methods, they do not contain the same number of data (see Fig. 20). Currently, the NASA ACOS product has the largest number of valid data points, followed by the new GOSAT-FOCAL product with about 20% less data.

### 5.1   Direct comparisons

There are enough common measurement points included in the different GOSAT products to perform a direct comparison.
Figure 21 shows exemplary a comparison between the GOSAT-FOCAL data for the year 2018 with the corresponding ACOS, BESD, SRON, NIES and UoL products. For each plot we only use data where both data sets have a valid $XCO_2$ value. For these data we performed a linear regression using the Orthogonal Distance Regression (ODR) method (see e.g. Boggs et al., 1987). Unlike common linear regression, ODR considers uncertainties for both axes (data sets) by minimising the orthogonal distances between the model curve and the data points. The ODR results are shown by the red line and its label. Number of
collocations and median/mean and standard deviations of the differences are given in the titles.

   Overall, the data scatter around the 1:1 line in a similar way for all comparisons. ODR slopes vary between the data sets from 0.84 (for FOCAL vs. ACOS) up to 1.08 (for FOCAL vs. BESD). Most collocations are available for the ACOS data set because this has the largest number of valid data. Mean and median differences are quite similar and reach from -0.17 ppm (comparison to BESD) to 0.67 ppm (comparison with UoL). The scatter (standard deviation of the differences) reaches from
1.4 ppm (ACOS, NIES) to 1.8 ppm (BESD).

### 5.2   TCCON comparisons

The TCCON network provides high-quality $XCO_2$ (and other) data which are currently considered to be the main reference for greenhouse gas data obtained from satellite measurements. Therefore we compared the different GOSAT data sets with collocated TCCON measurements from 2009 to the end of 2018. BESD data are not included, because they do not cover the
complete time interval. Collocation criteria are:

- Maximum time difference of 2 h.

- Maximum spatial distance of satellite measurement from TCCON station 500 km.

- Maximum surface elevation difference between satellite measurement and TCCON station 250 m.





In addition to these criteria we also consider in the validation only stations / TCCON data sets, which have at least 50

collocations for all algorithms. This improves the comparability of regional and seasonal biases. As a consequence, not all stations listed in Tab. 1 contribute to the validation.

The comparison procedure is the same as used by Reuter et al. (2020) and described by Reuter et al. (version 3.1, 03-11-2019). In summary, for each TCCON site, the time series of satellite minus TCCON differences are computed under consideration of the averaging kernels, i.e. different vertical sensitivities. The resulting time series are fitted with a trend

model, which includes an offset term, a slope term, and a sine term for seasonal fluctuations. The offset term is considered the station bias and the station scatter is computed from the standard deviation of the fit residual. Results for the time series at the TCCON stations are shown in Fig. 22. Overall, the temporal variations of $XCO_2$ are well reproduced by all data.

Figure 23 shows as a summary of the TCCON comparisons the derived bias and scatter for the different stations and products. The new GOSAT-FOCAL product compares well with the other data sets. Its differences to TCCON have a station to station

bias (the standard deviation of the station bias) of 0.56 ppm and a mean scatter (RMS scatter per station) of 1.89 ppm. The seasonal component of the bias has a station to station average standard deviation of 0.37 ppm. Overall, the ACOS product performs best in this comparison.

Note that the biases shown in Fig. 23 correspond essentially to a bias anomaly since a global bias was removed from all products. Therefore different signs of biases for different products could be coincidental. However, the biases of FOCAL and

ACOS are consistent with the biases found by Reuter et al. (2020).

Via the TCCON comparison it is also possible to validate the reported precision of the FOCAL data products (i.e. the specified $XCO_2$ error). The basic idea is to estimate the "true" precision from the variability of the $XCO_2$ bias relative to trend-corrected, collocated TCCON data. For this purpose, we define 20 bins with increasing reported $XCO_2$ uncertainty and compute the corresponding true precision from the scatter relative to TCCON (i.e., the fit residual mentioned above).

The corresponding scatter plot is shown in Fig. 24. We use the fitted linear curve to correct the reported uncertainty of the GOSAT-FOCAL data. After the correction, all data scatter around the 1:1 line (dashed).

A similar correction will be performed for the GOSAT-2 FOCAL product as soon as sufficient data (GOSAT-2 as well as TCCON) are available, which is currently not yet the case.

### 5.3 Time series

To investigate the temporal behaviour of the FOCAL $XCO_2$ data sets, we performed comparisons based on monthly data from 2009 to 2019, which were spatially gridded to $5° \times 5°$ (examples are shown in Figs. 18 and 19). Similar data sets have been generated for the SRON, UoL, ACOS and NIES GOSAT products. We also produced a corresponding gridded GOSAT-BESD data set; since these are near-real-time (NRT) data only, there are no GOSAT BESD data before 2014 available (when the NRT processing started). GOSAT-2 data start in February 2019.

We then selected for each combination of GOSAT-FOCAL $XCO_2$ and a correlative data set grid points where the standard error of the mean is less than 1.6 ppm (as a basic quality filter, similar as done by Reuter et al., 2020). These data were then averaged over different latitudinal ranges, namely:





- Global (90°S – 90°N)

- Northern hemisphere (25°N – 90°N)

- Tropics (25°S – 25°N)

- Southern hemisphere (25°S – 90°S)

Figure 25 shows the results of these comparisons. The left plots display time series of the different data sets, the right plots the difference between the GOSAT-FOCAL $XCO_2$ and the reference data. All data products reproduce the overall increase of $XCO_2$ with time as well as the seasonal variations. On average, FOCAL data are typically about 0.5 ppm higher than the
465 other data sets (except for BESD). This is most likely related to the choice of the "true" $XCO_2$ for the bias correction. There are long-term changes in the order of 1 ppm over the complete time series, which differ for each data set. For example, the GOSAT-FOCAL data show in the tropics relative to SRON a higher value at the start of the time series, but both data sets agree at the end. On the other hand, the average difference to the UoL data in the northern hemisphere is negative during the first years, but increases to an almost constant small positive offset below about 0.5 ppm. There is not much difference in the
470 temporal behaviour between the GOSAT-FOCAL and the ACOS and NIES time series. The seasonal shapes also differ slightly with amplitudes of about 0.5 ppm with somewhat larger differences in the southern hemisphere where seasonal variations are generally smaller.

Overall, the agreement within the GOSAT data sets is broadly consistent with the systematic regional and seasonal biases derived from the TCCON validation, especially considering that all gridded data sets are based on a different spatial and
475 temporal sampling. Also, the FOCAL products for GOSAT and GOSAT-2 seem to agree quite well, but more GOSAT-2 data is needed to confirm this.

## 6 Conclusions

Based on the FOCAL retrieval method a new $XCO_2$ data set for GOSAT and a first $XCO_2$ data set for GOSAT-2 have been generated, making use of both measured polarisation directions. The GOSAT-FOCAL data set compares well with corresponding
data from other currently available GOSAT retrieval algorithms, i.e. the RemoTec product from SRON, the UoL FP product, the NASA ACOS product, the NIES product and the BESD product from IUP. All data sets use different filtering and bias correction schemes and therefore comprise also a different number and sampling of data. The GOSAT-FOCAL product performs well in this context and has almost as many valid data as the ACOS product. Based on gridded data, differences in long-term variations of all data sets in the order of 1 ppm per decade are observed. Also, seasonal variations differ by about 0.5 ppm.
Comparisons with ground-based TCCON data reveal for the GOSAT-FOCAL product an overall station to station bias of 0.56 ppm and a mean scatter of 1.89 ppm. These values are comparable to and in some cases even better than those of the already existing GOSAT products of which some have less valid data.

The first GOSAT-2 results using the FOCAL method are also quite promising, but further investigations, longer time series and more correlative data sets are required for a quantitative assessment of the GOSAT-2-FOCAL data quality.





Overall, the FOCAL method has proven to be computationally fast and to produce $XCO_2$ results with similar accuracy as other, typically more time consuming, retrieval algorithms. This is the case not only when applied to OCO-2 but also for GOSAT and GOSAT-2. FOCAL is therefore considered to be a good candidate algorithm for future satellite sensors producing large amounts of data, like the forthcoming European anthropogenic $CO_2$ Monitoring (CO2M) mission.

*Data availability.*  The GOSAT-FOCAL V1.0 data set and the preliminary GOSAT-2-FOCAL data are available on request from the authors.

*Author contributions.*  S. Noël adapted the FOCAL method to GOSAT and GOSAT-2, generated the FOCAL data products and performed the validation. M. Reuter developed the FOCAL method and provided the "true" databases and the TCCON validation tools. J. Borchardt provided the used python implementation for the SC4C methane climatology from O. Scheising. M. Hilker provided the original python implementation of FOCAL (OCO-2 version). A. Di Noia and Y. Yoshida provided the UoL and NIES GOSAT data products. H. Suto provided information on GOSAT and GOSAT-2, especially the preliminary GOSAT-2 ILS.

The following co-authors provided TCCON data: M. Buschmann, N. M. Deutscher, D. G. Feist, D. W. T. Griffith, F. Hase, R. Kivi, I. Morino, J. Notholt, H. Ohyama, C. Petri, J. R. Podolske, D. F. Pollard, M. K. Sha, K. Shiomi, R. Sussmann, Y. Té, V. A. Velazco, T. Warneke.
All authors provided support in writing the paper.

*Competing interests.*  The authors declare that they have no conflict of interest.

*Acknowledgements.*  GOSAT and GOSAT-2 spectral data have been provided by JAXA and NIES. CarbonTracker CT2019 and CT-NRT.v2020-

1 results were provided by NOAA ESRL, Boulder, Colorado, USA from the website at http://carbontracker.noaa.gov. ABSCO cross sections for $CO_2$ were provided by NASA and the ACOS/OCO-2 team. GMTED2010 topography data were provided by the U.S. Geological Survey (USGS) and the National Geospatial-Intelligence Agency (NGA). We thank the European Center for Medium Range Weather Forecasts (ECMWF) for providing us with analysed meteorological fields (ERA5 data).
We thank the OCO-2 Science Team for the GOSAT ACOS Level 2 $XCO_2$ product obtained from https://oco2.gesdisc.eosdis.nasa.gov/

data/GOSAT_TANSO_Level2/ACOS_L2_Lite_FP.9r/, https://doi.org/10.5067/VWSABTO7ZII4, last access: 16 October 2020. The SRON GOSAT $XCO_2$ data product has been obtained from the Copernicus Climate Data Store (https://cds.climate.copernicus.eu/, last assess: 15-Oct-2020).
We used TCCON GGG2014 ground-based validation data (see Tab. 1). TCCON data from the Eureka and Izaña stations were provided by K. Strong and O. Garcia, respectively. The Ascension Island TCCON station has been supported by the European Space Agency (ESA)

under grant 4000120088/17/I-EF and by the German Bundesministerium für Wirtschaft und Energie (BMWi) under grants 50EE1711C and 50EE1711E. We thank the ESA Ariane Tracking Station at North East Bay, Ascension Island, for hosting and local support. The TCCON site at Réunion Island is operated by the Royal Belgian Institute for Space Aeronomy with financial support since 2014 by the EU project ICOS-Inwire and the ministerial decree for ICOS (FR/35/IC1 to FR/35/C5) and local activities supported by LACy/UMR8105 – Université



de La Réunion. The Paris TCCON site has received funding from Sorbonne Université, the French research center CNRS, the French space
agency CNES, and Région Île-de-France. The TCCON stations at Rikubetsu, Tsukuba and Burgos are supported in part by the GOSAT series
project. Local support for Burgos is provided by the Energy Development Corporation (EDC, Philippines). N. M. Deutscher is funded by
ARC Future Fellowship FT180100327. Darwin and Wollongong TCCON stations are supported by ARC grants DP160100598, LE0668470,
DP140101552, DP110103118 and DP0879468, and Darwin receives additional support from NASA grants NAG5-12247 and NNG05-
GD07G and technical assistance from the Australian Bureau of Meteorology. The TCCON stations Garmisch and Zugspitze have been
supported by the European Space Agency (ESA) under grant 4000120088/17/I-EF and by the German Bundesministerium für Wirtschaft
und Energie (BMWi) via the DLR under grant 50EE1711D as well as by the Helmholtz Society via the research program ATMO.

This work has received funding from JAXA (GOSAT and GOSAT-2 support, contracts 19RT000692 and JX-PSPC-527269), EUMETSAT
(FOCAL-CO2M study, contract EUM/CO/19/4600002372/RL), ESA (GHG-CCI+ project, contract 4000126450/19/I-NB) and the State and
the University of Bremen.



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

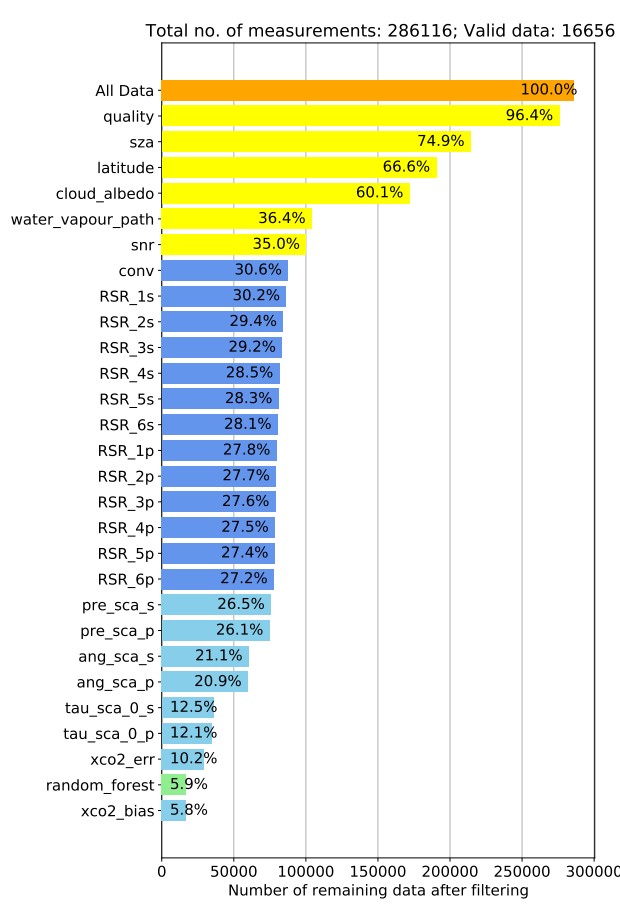

**Figure 1.** Example for GOSAT data filtering during the different processing steps (April 2019). Filters are listed in sequential order from top to bottom on the vertical axis. Numbers in the horizontal bars denote the percentage of remaining data after this filter was applied. Orange: Total number of measurements before filtering. Yellow: Pre-processing filters. Blue: Step 1 post-processing filters (convergence and noise). Green: Random forest post-processing filter. Light blue: Additional post-processing filters.





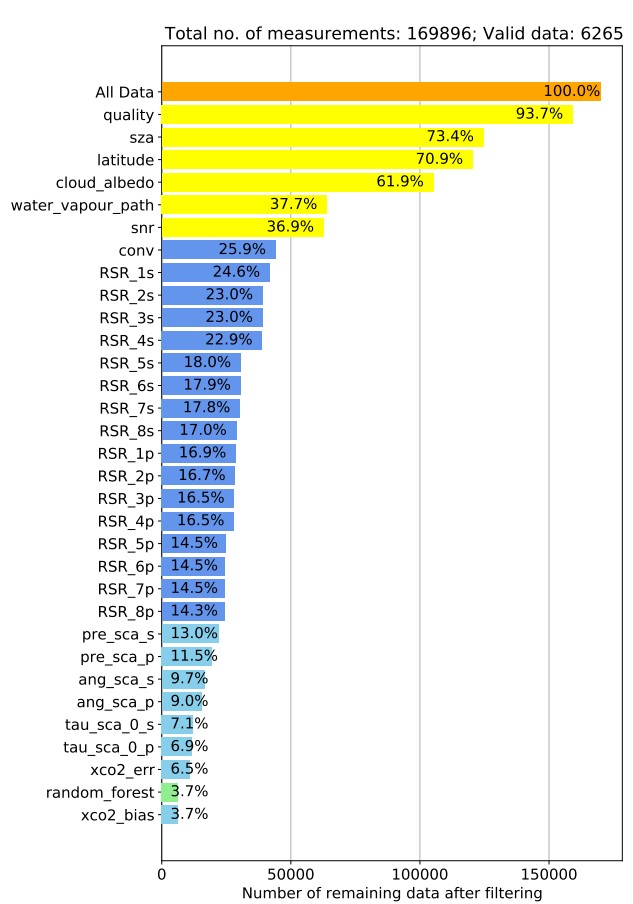

**Figure 2.** Same as Fig. 1, but for GOSAT-2.



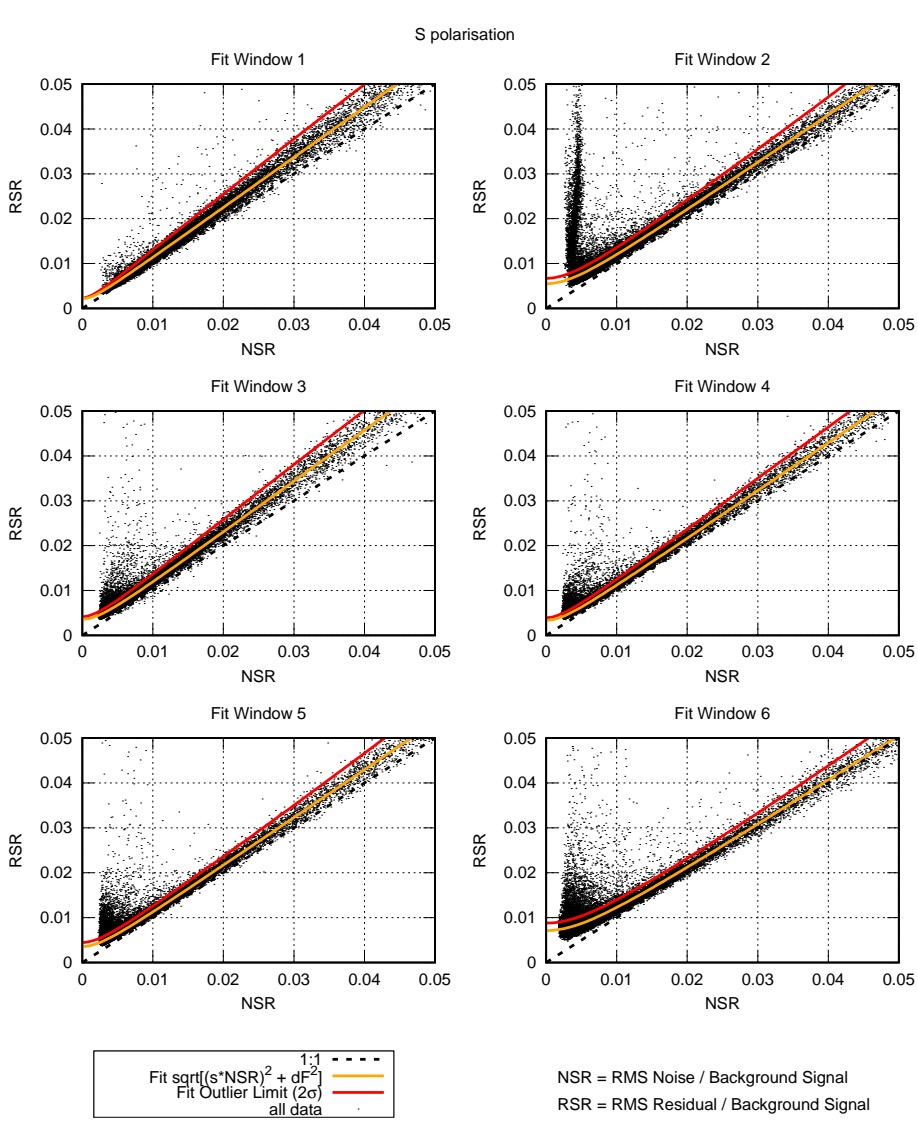

**Figure 3.** GOSAT noise model (S polarisation).





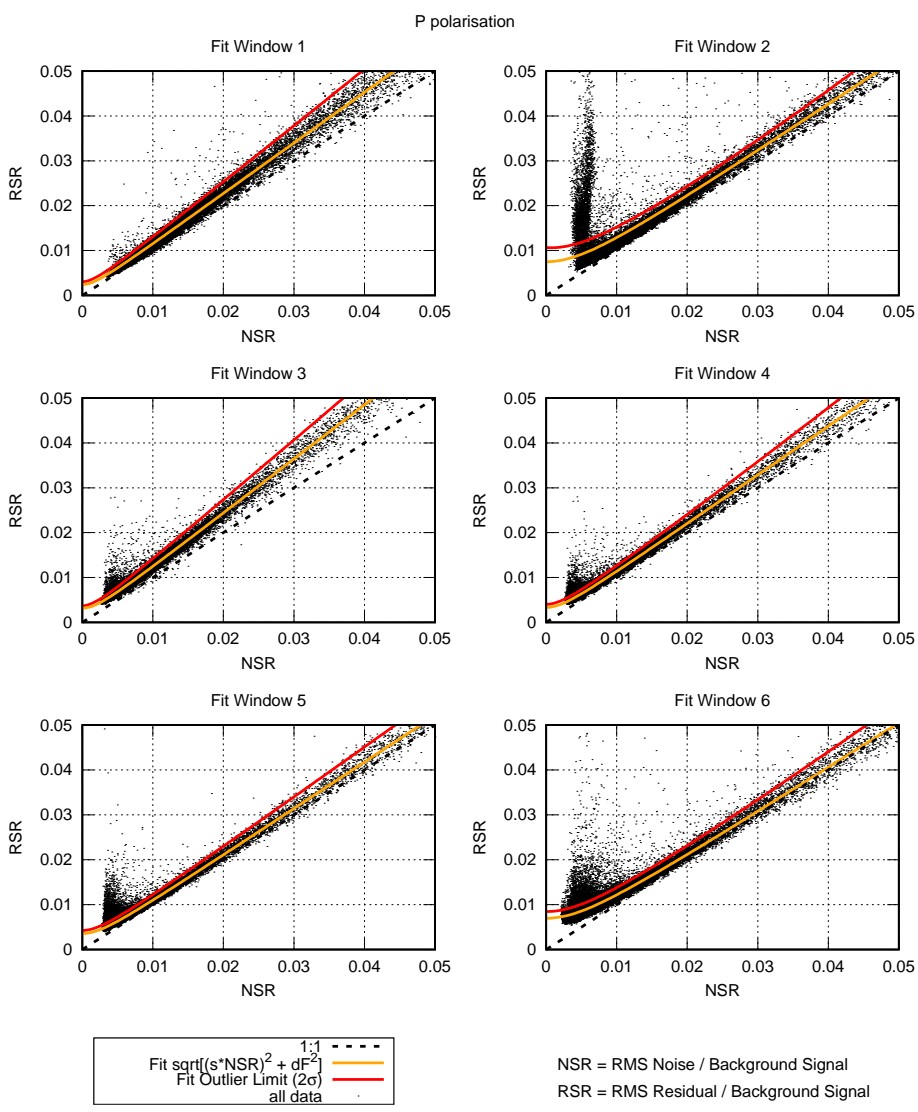

**Figure 4.** GOSAT noise model (P polarisation).





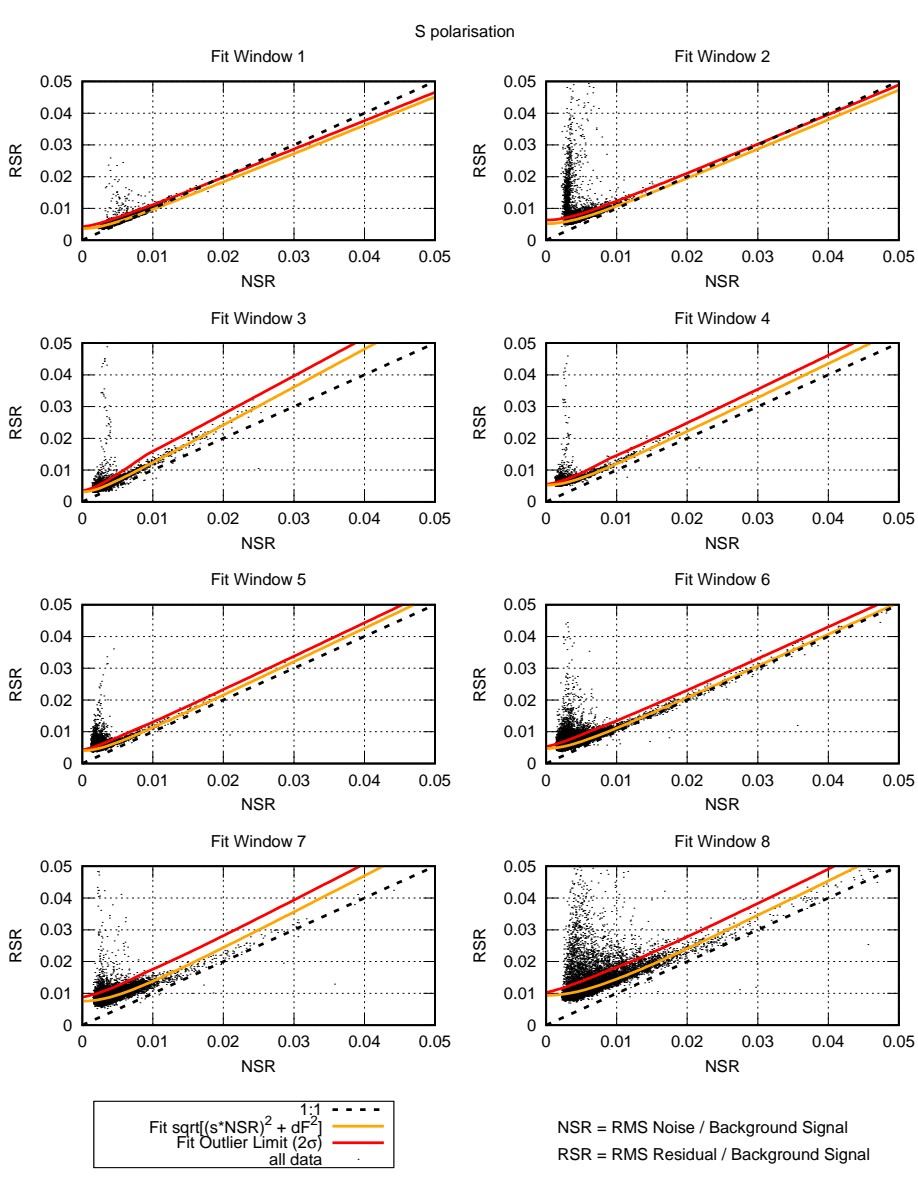

**Figure 5.** GOSAT-2 noise model (S polarisation).





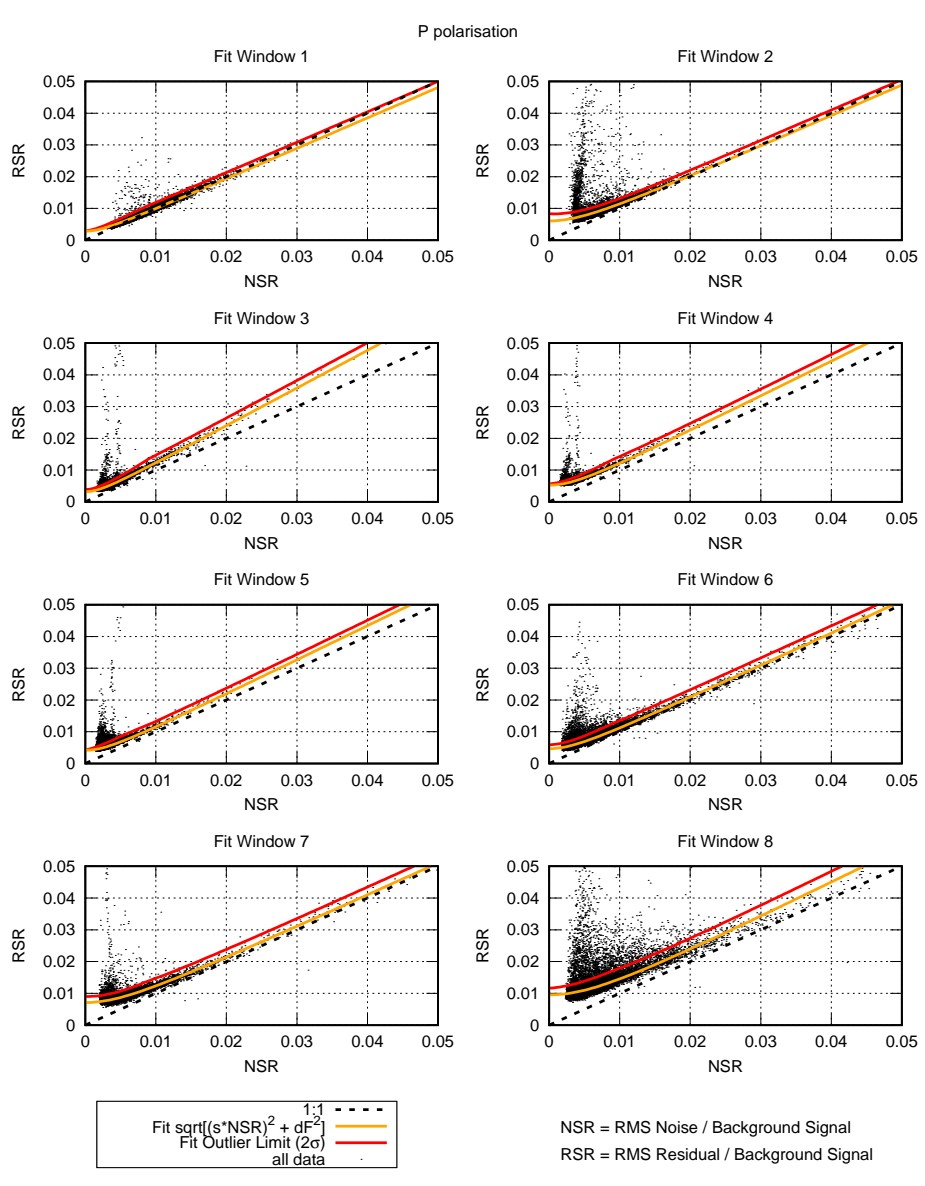

**Figure 6.** GOSAT-2 noise model (P polarisation).





**Figure 7.** Random forest results for GOSAT. Left (a–c): Results from random forest filter. Right (d–f): Results from random forest bias correction. Top: Normalised relevance (score) of all filter variables. Middle/bottom: Selected variables and their relevance for land/water surface.




**Figure 8.** As Fig. 7, but for GOSAT-2.





**Figure 9.** Example for a single GOSAT measurement (S polarisation): Measured (red) and retrieved (green) spectra in the different fit windows; because of the good agreement the red curve is essentially barely visible below the green curve. Radiance unit is $\mathrm{W/cm^2/cm^{-1}/sr}$.





**Figure 10.** Same as Fig. 9 but for P polarisation.



**Figure 11.** Same as Fig. 9 but for GOSAT-2 with two additional fit windows.

**Figure 12.** Same as Fig. 11 but for P polarisation.



**Figure 13.** GOSAT residuals (measurement - fit) for data from Fig. 9 (S polarisation). Green: Unsmoothed. Blue: Smoothed with a boxcar of width 21 spectral pixels (= 4.2 cm$^{-1}$). Red: Estimated noise error range. Radiance difference unit is W/cm$^2$/cm$^{-1}$/sr.





**Figure 14.** Same as Fig. 13 but for P polarisation.







**Figure 15.** Same as Fig. 13 but for GOSAT-2.



**Figure 16.** Same as Fig. 14 but for GOSAT-2.





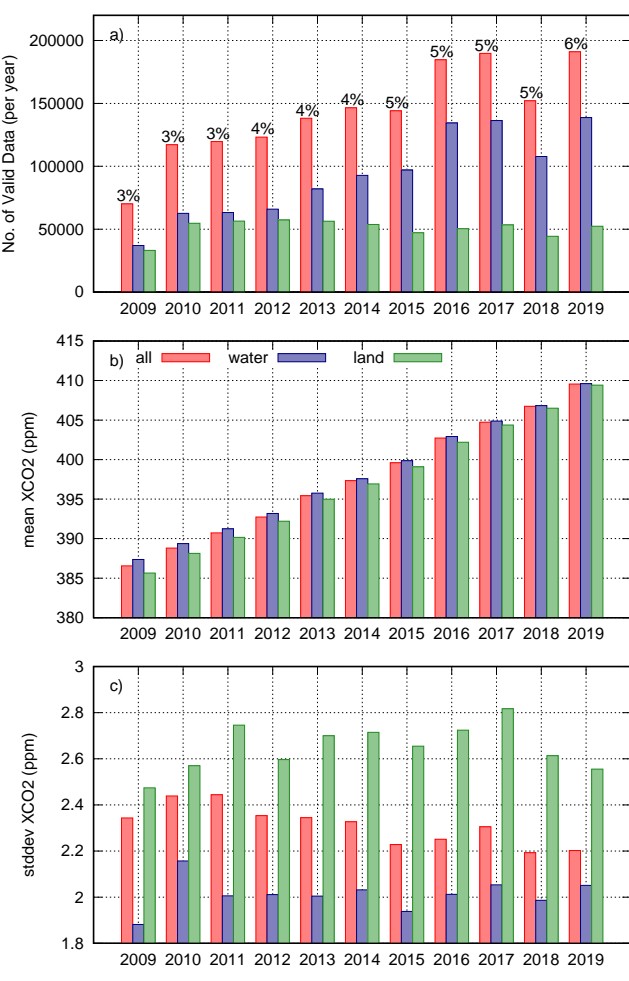

**Figure 17.** Statistics for valid GOSAT measurements (after pre- and post-processing filtering) for each year. Blue: Measurements over water. Green: Measurements over land. Red: All data. a) Number of valid measurements, incl. percentage of all originally available measurements. b) Global mean $XCO_2$. c) Corresponding standard deviation.





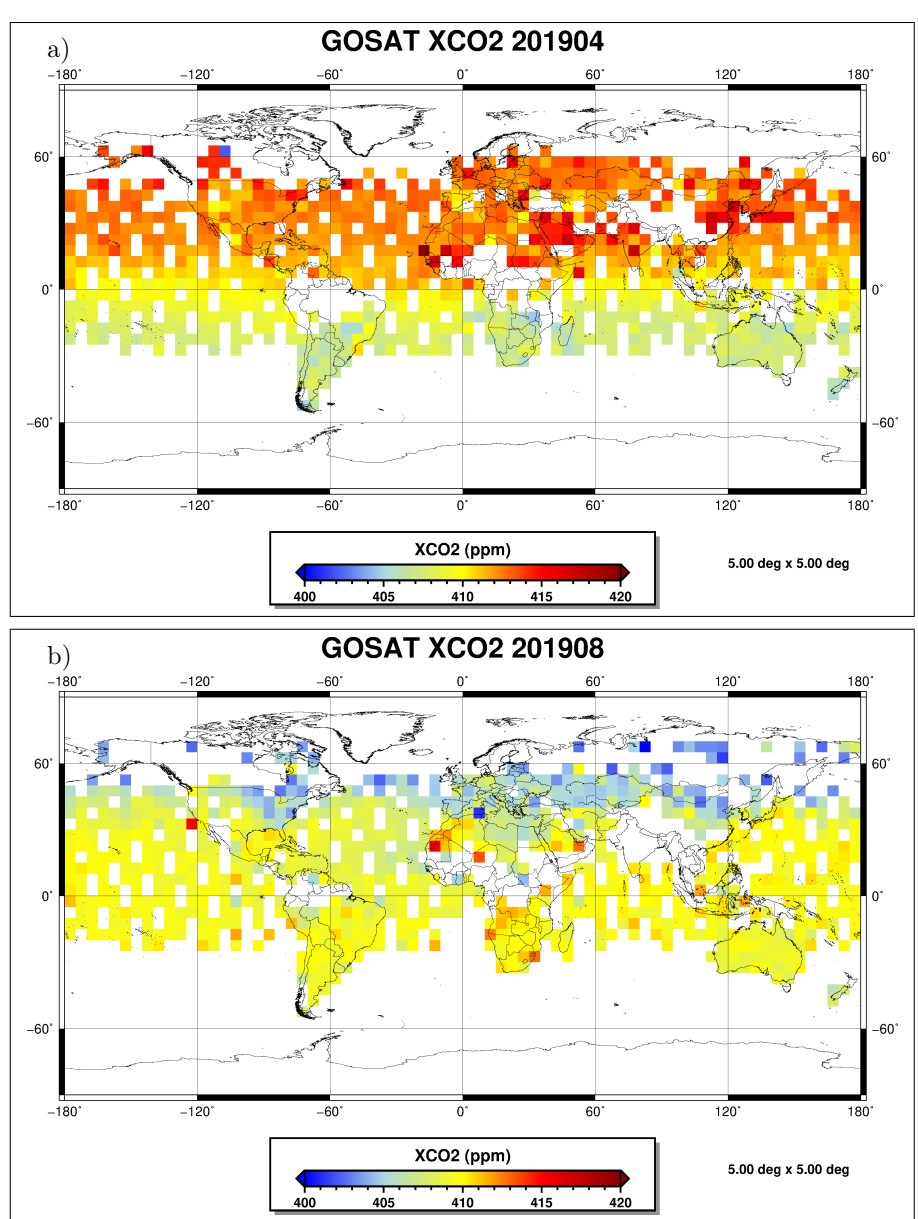

**Figure 18.** Example for gridded GOSAT $XCO_2$ data. a) April 2019. b) August 2019.

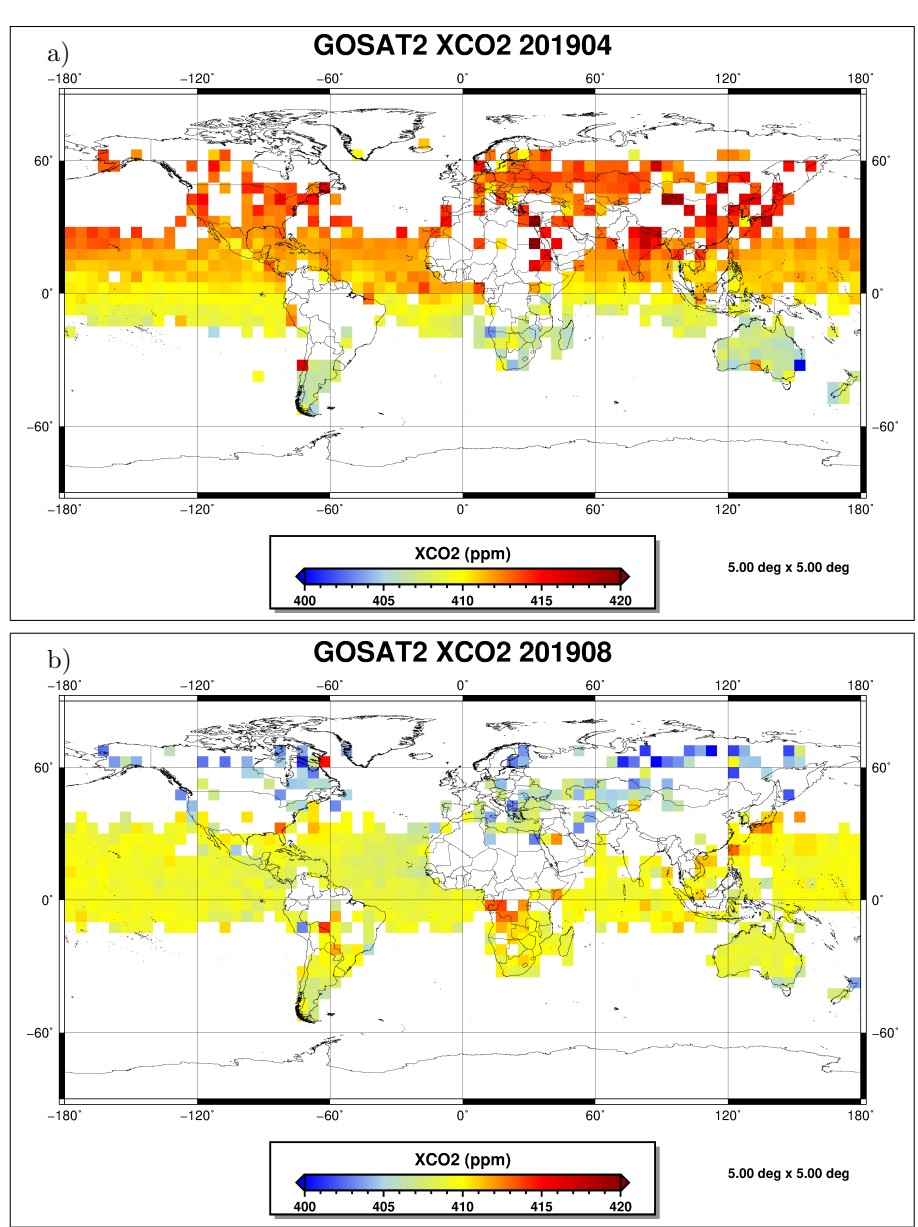

**Figure 19.** Example for gridded GOSAT-2 $XCO_2$ data. a) April 2019. b) August 2019.





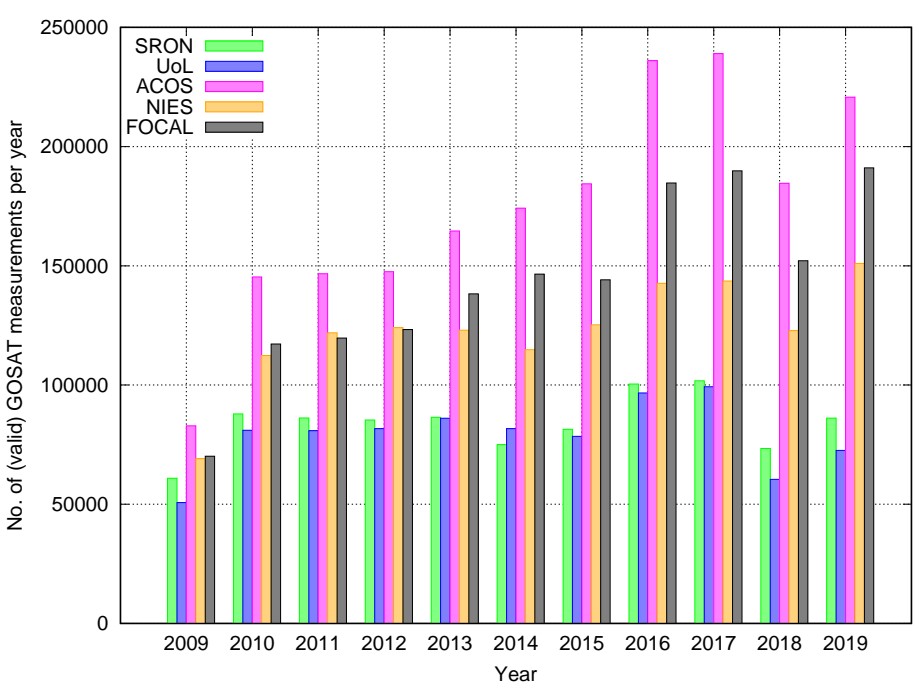

**Figure 20.** Number of valid $XCO_2$ data points in the different GOSAT products from 2009 to 2019.

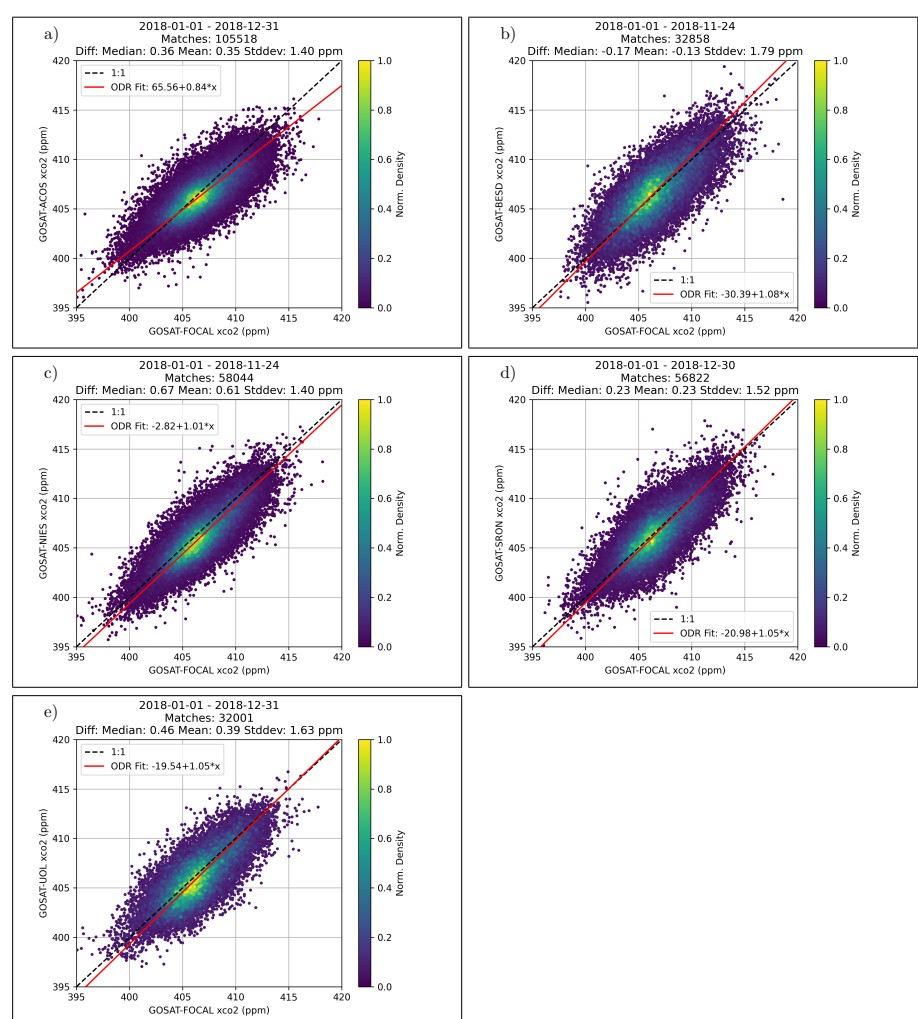

**Figure 21.** Comparison of GOSAT-FOCAL data (x axis) from 2018 with other GOSAT data (y axis). The colour of the data points corresponds to the density of data points at that location (normalised to a maximum value of 1). The dashed line corresponds to perfect agreement. The red line shows the result of a linear fit using the Orthogonal Distance Regression (ODR) method. The total number of collocated data as well as median, mean and standard deviation of the $XCO_2$ differences between the two data sets are given in the title of the sub-plots. a) FOCAL vs. ACOS b) FOCAL vs. BESD. c) FOCAL vs. NIES. d) FOCAL vs. SRON. e) FOCAL vs. UoL.





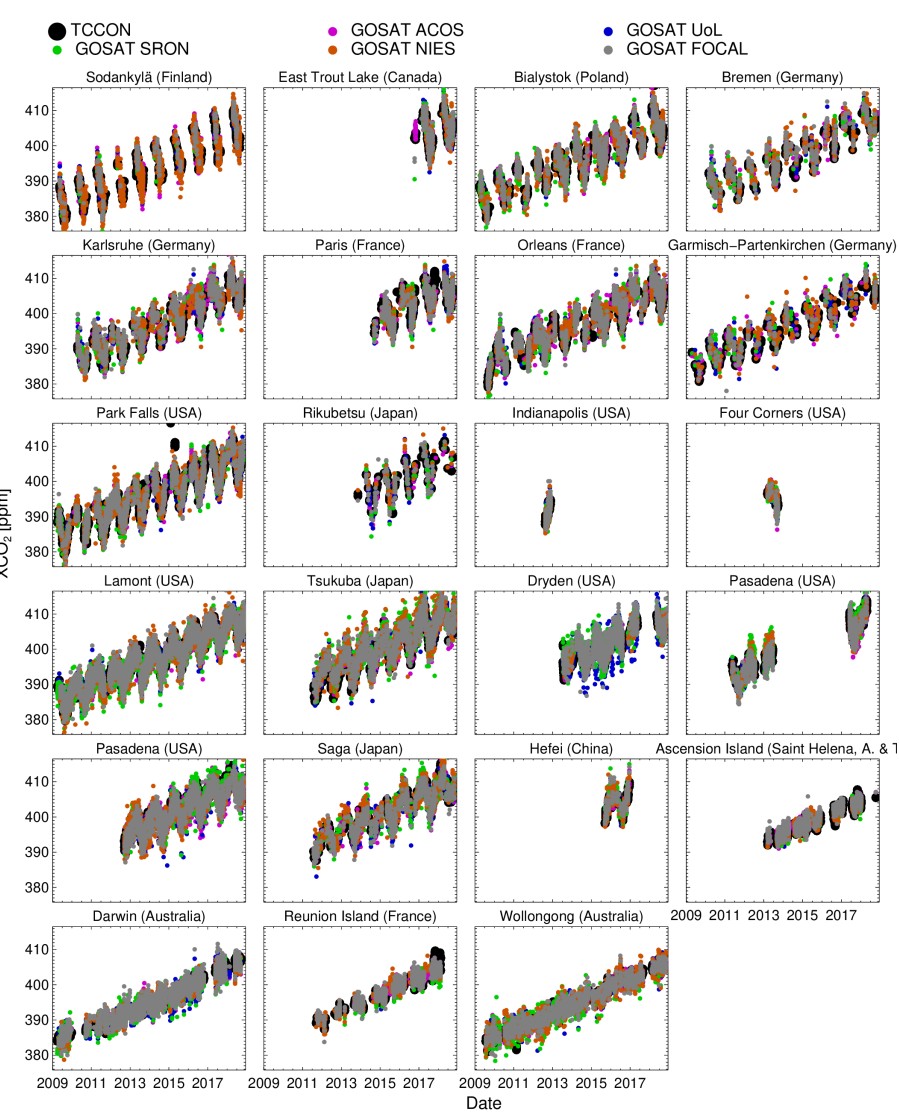

**Figure 22.** Time series of collocated GOSAT data at various TCCON sites for different data products including the new GOSAT-FOCAL data set.





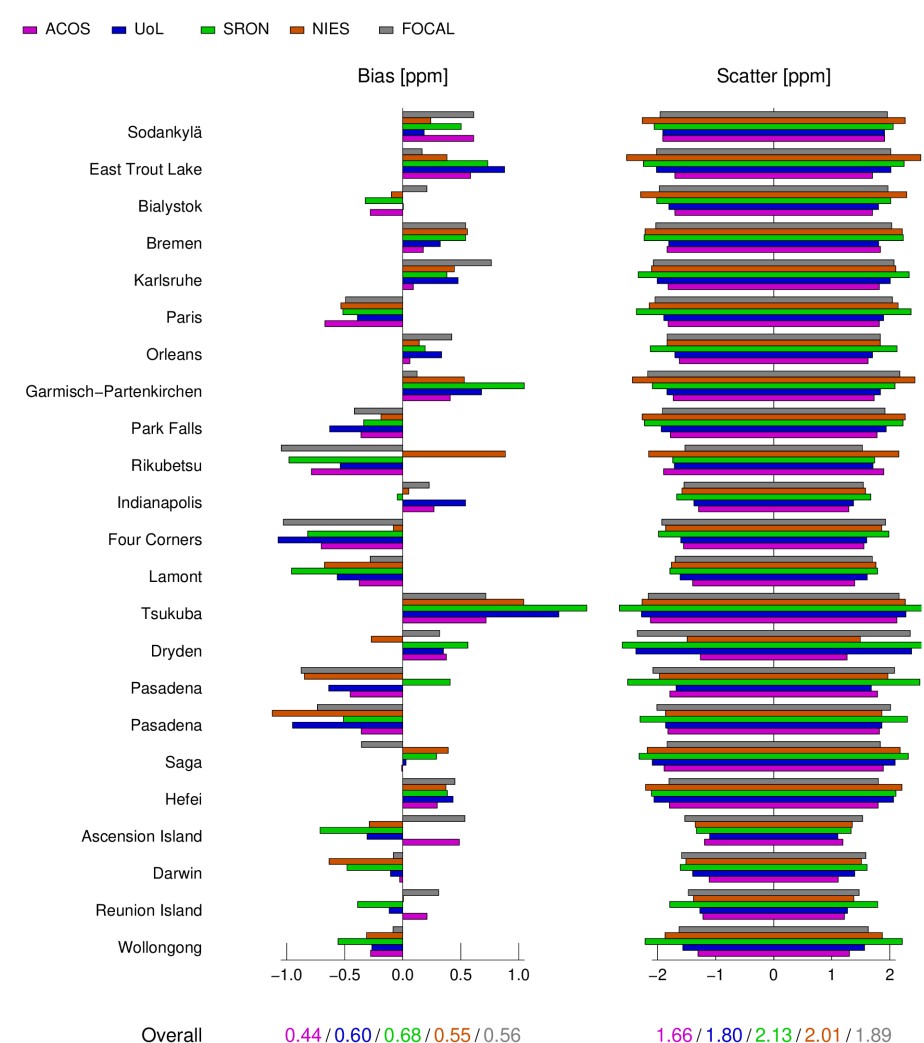

**Figure 23.** Overview of TCCON validation results.



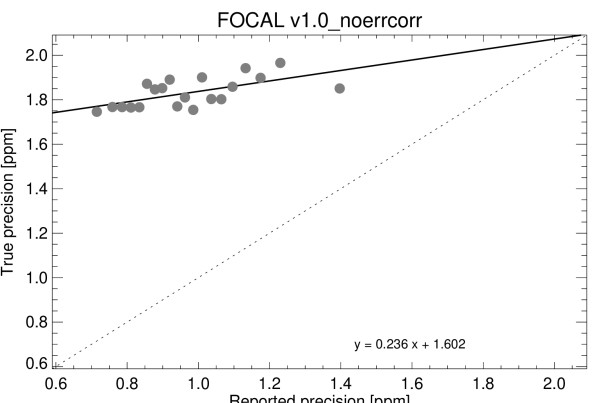

**Figure 24.** Comparisons of the (binned) original $XCO_2$ errors (reported precision without correction) of the GOSAT-FOCAL product with estimated errors based on collocated TCCON data (true precision).







**Figure 25.** Gridded monthly mean time series of different GOSAT $XCO_2$ products. Left: Time series of mean $XCO_2$ for four different regions (from top to bottom: Global, northern hemisphere, tropics, southern hemisphere). Right: Corresponding differences to the GOSAT-FOCAL product.



**Table 1.** TCCON stations used in this study.

| Site | Lon. (deg) | Lat. (deg) | Elev. (km) | Reference(s) |
|---|---|---|---|---|
| Ascension Island (SH) | -14.33 | -7.92 | 0.01 | Feist et al. (2014) |
| Bialystok (PL) | 23.03 | 53.23 | 0.18 | Deutscher et al. (2019) |
| Bremen (DE) | 8.85 | 53.10 | 0.04 | Notholt et al. (2019a) |
| Burgos (PH) | 120.65 | 18.53 | 0.04 | Morino et al. (2018b) |
| Darwin (AU) | 130.89 | -12.42 | 0.03 | Griffith et al. (2014a) |
| Edwards (US) | -117.88 | 34.96 | 0.70 | Iraci et al. (2016a) |
| East Trout Lake (CA) | -104.99 | 54.35 | 0.50 | Wunch et al. (2017) |
| Eureka (CA) | -86.42 | 80.05 | 0.61 | Strong et al. (2019) |
| Four Corners (US) | -108.48 | 36.80 | 1.64 | Dubey et al. (2014) |
| Garmisch-Partenkirchen (DE) | 11.06 | 47.48 | 0.74 | Sussmann and Rettinger (2018a) |
| Hefei (CN) | 117.17 | 31.90 | 0.04 | Liu et al. (2018) |
| Indianapolis (US) | -86.00 | 39.86 | 0.27 | Iraci et al. (2016b) |
| Izaña (ES) | -16.50 | 28.30 | 2.37 | Blumenstock et al. (2017) |
| Karlsruhe (DE) | 8.43 | 49.10 | 0.11 | Hase et al. (2014) |
| Lamont (US) | -97.49 | 36.60 | 0.32 | Wennberg et al. (2016) |
| Lauder (NZ) | 169.68 | -45.04 | 0.37 | Sherlock et al. (2014a, b) |
| | | | | Pollard et al. (2019) |
| Ny Ålesund (NO) | 11.90 | 78.90 | 0.02 | Notholt et al. (2019b) |
| Orleans (FR) | 2.11 | 47.97 | 0.13 | Warneke et al. (2019) |
| Paris (FR) | 2.36 | 48.85 | 0.06 | Te et al. (2014) |
| Park Falls (US) | -90.27 | 45.95 | 0.44 | Wennberg et al. (2017) |
| Pasadena (US) | -118.13 | 34.13 | 0.21 | Wennberg et al. (2014) |
| Reunion Island (FR) | 55.49 | -20.90 | 0.09 | De Mazière et al. (2017) |
| Rikubetsu (JP) | 143.77 | 43.46 | 0.36 | Morino et al. (2017) |
| Saga (JP) | 130.29 | 33.24 | 0.01 | Kawakami et al. (2014) |
| Sodankylä (FI) | 26.63 | 67.37 | 0.18 | Kivi et al. (2014) |
| Tsukuba (JP) | 140.12 | 36.05 | 0.03 | Morino et al. (2018a) |
| Wollongong (AU) | 150.88 | -34.41 | 0.03 | Griffith et al. (2014b) |
| Zugspitze (DE) | 10.98 | 47.42 | 2.96 | Sussmann and Rettinger (2018b) |





**Table 2.** Pre-processing filter limits.

| Filter | Value |
|---|---|
| GOSAT quality flag | "good" or "poor" |
| GOSAT-2 quality flag | "good", "fair" or "poor" |
| Maximum solar zenith angle | 70° |
| Maximum latitude | 70° |
| Minimum SNR | 10 |
| Maximum cloud albedo | 0.7 |
| Maximum water vapour path | 4.0 |





**Table 3.** Parameters for cloud filtering.

| Cloud Albedo | | |
|---|---|---|
| GOSAT Band | Waveno. Range ($\mathrm{cm}^{-1}$) | Irradiance ($\mathrm{W/cm^2/s/cm^{-1}}$) |
| SWIR-1 | 13190–13210 | 7.4e-6 |
| SWIR-2 | 6267–6279 | 6.0e-6 |
| SWIR-3 | 4800–4810 | 4.3e-6 |
| Water Vapour Path | | |
| GOSAT Band | Waveno. Range ($\mathrm{cm}^{-1}$) | |
| SWIR-3 | 5176–5193 | |





**Table 4.** Definition of GOSAT/GOSAT-2 spectral fit windows (same for S and P). Windows 7 and 8 are only available for GOSAT-2.

| No. | Primary target | Waveno. range ($cm^{-1}$) | Considered gases |
|---|---|---|---|
| 1 | SIF | 13170 – 13220 | $O_2$ |
| 2 | $O_2$ | 12930 – 13170 | $O_2$ |
| 3 | HDO | 6337 – 6410 | $CO_2$, $H_2O$, HDO, $CH_4$ |
| 4 | $CO_2$ | 6161 – 6297 | $CO_2$, $H_2O$, HDO, $CH_4$ |
| 5 | $CH_4$ | 5945 – 6135 | $CO_2$, $H_2O$, HDO, $CH_4$ |
| 6 | $CO_2$ | 4801 – 4907 | $CO_2$, $H_2O$, HDO |
| 7 | $N_2O$ | 4364 – 4449 | $N_2O$, $H_2O$, HDO, $CH_4$ |
| 8 | CO | 4228 – 4328 | CO , $H_2O$, HDO, $CH_4$ |



**Table 5.** State vector elements and related retrieval settings. A-priori values are also used as first guess. "Fit windows" lists the spectral windows (see Tab. 4) from which the element is determined. "all" means that an element is determined from all fit windows of the specified polarisation. "each" means that a corresponding element is fitted in each fit window. A-priori values labelled as "PP" are taken from pre-processing; "est." denotes that they have been estimated from the background signal

| Element | Fit windows | A-priori | A-priori uncertainty | Comment |
|---|---|---|---|---|
| | | Gases | | |
| co2_lay | 3,4,5,6 (S&P) | PP | 10.0 | $CO_2$ profile (5 layers), in ppm |
| ch4_lay | 3,4,5 (S&P) | PP | 0.045 | $CH_4$ profile (5 layers), in ppm |
| h2o_lay | 3,4,5,6 (S&P) | PP | 5.0 | $H_2O$ profile (5 layers), in ppm |
| sif_fac | 1 (S&P) | 0. | 5. | SIF spectrum scaling factor |
| delta_d | 3,4,5,6 (S&P) | -200. | 1000. | $\delta D$ profile scaling factor |
| n2o_scl | 7 (S&P) | 1. | 0.1 | $N_2O$ profile scaling factor, only GOSAT-2 |
| co_scl | 8 (S&P) | 1. | 1.0 | CO profile scaling factor, only GOSAT-2 |
| | | Scattering parameters | | |
| pre_sca_s | all S | 0.2 | 1. | Layer height (pressure), S |
| tau_sca_0_s | all S | 0.01 | 0.1 | Optical depth, S |
| ang_sca_s | all S | 4.0 | 1. | Ångström coefficient, S |
| pre_sca_p | all P | 0.2 | 1. | Layer height (pressure), P |
| tau_sca_0_p | all P | 0.01 | 0.1 | Optical depth, P |
| ang_sca_p | all P | 4.0 | 1. | Ångström coefficient, P |
| | | Polynomial coefficients (surface albedo) | | |
| poly0 | each | est. | 0.01 | estimated surface albedo |
| poly1 | each | 0.2 | 0.1 | |
| poly2 | each | 0.1 | 0.1 | not in SIF window (1) |
| | | Spectral corrections | | |
| wav_shi | each | 0.0 | 0.1 | Wavenumber shift |
| wav_squ | each | 0.0 | 0.001 | Wavenumber squeeze |





**Table 6.** Parameters of GOSAT noise model.

| Fit window | $s$ | $\delta F$ | $a_0$ | $a_1$ | $a_2$ | NSR range |
|---|---|---|---|---|---|---|
| | | | S polarisation | | | |
| 1 | 1.12 | 2.17e-03 | 9.369e-05 | 1.613e-01 | -9.185e-01 | 0.003–0.049 |
| 2 | 1.07 | 5.50e-03 | 1.183e-03 | 2.557e-02 | 1.107e+00 | 0.003–0.047 |
| 3 | 1.14 | 3.61e-03 | 5.241e-04 | 1.251e-01 | -6.776e-01 | 0.003–0.043 |
| 4 | 1.07 | 3.37e-03 | 5.480e-04 | 7.250e-02 | 3.716e-02 | 0.003–0.041 |
| 5 | 1.07 | 3.58e-03 | 8.836e-04 | 3.433e-02 | 8.853e-01 | 0.003–0.047 |
| 6 | 1.00 | 7.12e-03 | 1.680e-03 | -9.060e-03 | 1.190e+00 | 0.001–0.047 |
| | | | P polarisation | | | |
| 1 | 1.13 | 2.42e-03 | 5.961e-04 | 8.736e-02 | 6.867e-01 | 0.003–0.049 |
| 2 | 1.05 | 7.50e-03 | 3.177e-03 | -1.109e-01 | 2.711e+00 | 0.003–0.049 |
| 3 | 1.21 | 3.17e-03 | 4.909e-04 | 1.226e-01 | 5.539e-02 | 0.003–0.037 |
| 4 | 1.09 | 3.33e-03 | 6.725e-04 | 4.661e-02 | 1.002e+00 | 0.003–0.035 |
| 5 | 1.04 | 3.58e-03 | 6.457e-04 | 5.710e-02 | 2.663e-01 | 0.003–0.039 |
| 6 | 1.00 | 6.96e-03 | 1.488e-03 | -9.996e-04 | 1.262e+00 | 0.003–0.049 |



**Table 7.** Parameters of GOSAT-2 noise model.

| Fit window | $s$ | $\delta F$ | $a_0$ | $a_1$ | $a_2$ | NSR range |
|---|---|---|---|---|---|---|
| S polarisation | | | | | | |
| 1 | 0.90 | 3.65e-03 | 5.895e-04 | 2.314e-01 | -1.524e+01 | 0.003–0.009 |
| 2 | 0.94 | 5.21e-03 | 1.143e-03 | 1.352e-02 | 4.058e+00 | 0.003–0.009 |
| 3 | 1.20 | 3.14e-03 | 3.972e-04 | 2.756e-01 | 7.753e+00 | 0.001–0.009 |
| 4 | 1.08 | 5.25e-03 | 2.491e-04 | 2.683e-01 | -4.440e-01 | 0.001–0.007 |
| 5 | 1.06 | 4.04e-03 | 2.184e-04 | 4.112e-01 | -2.730e+01 | 0.001–0.007 |
| 6 | 1.01 | 4.73e-03 | 6.001e-04 | 4.469e-01 | -2.822e+01 | 0.001–0.011 |
| 7 | 1.16 | 7.53e-03 | 1.289e-03 | 3.984e-01 | -1.377e+01 | 0.001–0.009 |
| 8 | 1.11 | 9.34e-03 | 9.305e-04 | 5.126e-01 | -2.263e+01 | 0.003–0.017 |
| P polarisation | | | | | | |
| 1 | 0.96 | 2.83e-03 | 8.010e-05 | 2.856e-01 | -9.105e+00 | 0.003–0.017 |
| 2 | 0.97 | 6.08e-03 | 2.258e-03 | -1.191e-01 | 6.365e+00 | 0.003–0.015 |
| 3 | 1.19 | 3.20e-03 | 6.571e-04 | 9.767e-02 | 1.049e+01 | 0.001–0.011 |
| 4 | 1.10 | 5.25e-03 | 3.868e-04 | 2.064e-01 | -1.886e+00 | 0.001–0.009 |
| 5 | 1.08 | 4.17e-03 | 6.688e-05 | 4.935e-01 | -3.445e+01 | 0.001–0.009 |
| 6 | 1.02 | 4.68e-03 | 1.181e-03 | 1.123e-01 | 1.245e+00 | 0.001–0.015 |
| 7 | 1.01 | 7.11e-03 | 1.907e-03 | -3.405e-02 | 1.012e+01 | 0.003–0.015 |
| 8 | 1.10 | 9.52e-03 | 2.093e-03 | 1.632e-01 | -2.418e+00 | 0.003–0.021 |





**Table 8.** Basic filter parameters.

| Filter | Range for valid data |
|---|---|
| Good convergence | $\chi^2 \leq 2$ |
| RSR (each fit window, S&P) | below outlier limit |
| Scattering layer height (S&P) | 0 to 1 |
| Ångström coefficient (S&P) | 1 to 5 |
| Scattering layer optical depth (S&P) | 0 to 0.02 |
| $XCO_2$ error | 0 to 2.0 ppm |



**Table 9.** Bias filter limits.

| Filter | Range for valid data |
| --- | --- |
| GOSAT land | -6.9 to -2.9 ppm |
| GOSAT water | -8.1 to -4.1 ppm |
| GOSAT-2 land | -4.0 to 0.0 ppm |
| GOSAT-2 water | -5.5 to -1.5 ppm |