# Peer review of "XCO2 retrieval for GOSAT and GOSAT-2 based on the FOCAL algorithm"

_Atmospheric Measurement Techniques, 2020_

## Referee Comment (RC1) · Anonymous Referee #2 · 29 Dec 2020

This paper describes and evaluates the FOCAL algorithm for the retrieval of XCO2 from GOSAT and GOSAT-2 measurements. It is said that the algorithm can retrieve other atmospheric parameters, but this is hardly described and the results are not evaluated. The paper can be of interest for the active community that develop retrieval algorithms for the spaceborne observations of atmospheric CO2. However, there are several parts that deserve better description as described below. Also, many figures are not useful or are poorly designed and deserve additional work. I therefore recommend this paper to be "accepted with major revision", accounting for the comments below.

Major comment Section 2.3 "True database" It remains unclear whether the actual TCCON measurements are used or not in the procedure. My current understanding is that the "valid" simulations are those where the total column, computed with an

homogeneous weighting and the same with the TCCON averaging kernel, differ by less than 0.75 ppm. If my understanding is correct, then (i) I do not understand why this is a valid criterium to select the most accurate simulation data and (ii) there is no need to compute the "daily mean" TCCON. If my understanding is incorrect, the description needs some re-writing. Line 138-139 "which were confirmed by TCCON" is really unclear, especially since it is said above that there is "more data in the Southern hemisphere" that is notably poor in TCCON coverage. How can such data be confirmed by TCCON. As a consequence, it is really unclear why the database build as described, can be considered as a truth

Section 3.1.2 Cloud Filter It is said that the Cloud Filter is based on the fact that clouds are bright (OK) and higher in the atmosphere so that there is little water vapour above them. Then, one my expect that, when the water vapor estimate is low, a cloud presence is suspected. Yet the description of the test indicates that a cloud is detected when the water vapor estimates is **larger** than a threshold. This is inconsistent.

Section 3.3.2 Random Forest Filter The difference between the estimate and the "true" reference database is used. It is said that the difference is subtracted by the global monthly mean bias. This assumes that the global mean bias of the reference is zero which is a strong assumption. This section lacks a quantitative discussion : What are the mean values of the differences to the "true" reference dataset. What is the order of magnitude of the bias correction ? Does it has some spatial patterns. In the case of ACOS, the bias correction is similar to the signal which is an important information. Is it the same here ? It is then said (line 321) that the random forest classification is accurate in about two thirds of the cases. How is this evaluated ? How can one decide whether it is accurate or not ?

Figures There is certainly no need for Figure 3 to 6. A couple of examples would be sufficient rather than the 28 pannels that indicate similar behaviors (and differences that are not commented) Figure 9 to 12 provide no usefull information Figure 13 to 16 could be limited to a few examples rather than the 30 pannels. I strongly recomment

to combine figure 9 and 13 so that one can identify whether the fine scale structures of the residuals correspond to absorption lines

Section 4; Results

This section contains several hypothetical statements "likely", "most likely", "which may explain". . . that deserve investigations

Other comment

Abstract : Line 21 "regional bias". There is no demonstration that TCCON is representative of a region, neither that the bias at the TCCON location is the same over a region

Section 2.2; line 111-112 Why a factor of 5 for $H_2O$ "to reduce dependencies on the a-priori" but not the same factor for the $CO_2$ Line 117 "very accurate". Please quantify Line 124 : "daily mean". I understand the mean is over 4 hours. How can this be considered a daily men ? Line 188 : "It is given by the ratio between the median radiance and the median of the estimated noise in this spectral range". Unclear Line 278 : The case is rejected when the Angstrom coefficient is outside of the range $[1 - 5]$. This is strange. Clouds and aerosols can have Anstrom coefficients that are close to zero. Conversely, values larger than 2 have never been reported to my knowledge. Line 335 "But with this filter applied". Which filter ? Line 343 : What is the order of magnitude of the bias correction ? Line 349 "On the derived XCO2 bias". What bias is that ? Is it before or after the correction ? The paragraph indicates it is after correction, but then how can it be evaluated ?

---

## Referee Comment (RC2) · Anonymous Referee #1 · 16 Feb 2021

Review of "XCO2 retrieval for GOSAT and GOSAT-2 based on the FOCAL algorithm" by Noel et al.

This paper describes applying the FOCAL retrieval of Max Reuter et al. (2017), originally developed for OCO-2, to GOSAT and GOSAT-2 data products. This is novel in and of itself, but particularly so for its application to GOSAT-2. The GOSAT XCO2 retrievals show good statistics when compared to TCCON as well as other established GOSAT XCO2 retrieval algorithms. Other gases are also retrieved (e.g., CH4) but are not discussed in this work. The FOCAL algorithm is particularly fast and therefore may serve as a candidate retrieval algorithm for the upcoming European CO2M mission.

**General Comments**

Overall, I found this paper useful and interesting, and will serve as an important reference. The subject matter is important, the layout of the paper is logical, the reasoning sound, and the results are generally laid out well. However, there are a number of problems that need to be addressed. While details of the retrieval, filtering, and bias correction were presented in a straightforward way, it was quite dry with little learned. Especially in the part about the random forest filter, which was used for both filtering and bias correction, but with little attempt on the part of the authors to explain the relevance of the features identified. The same goes for the prefilters, where it appeared that thresholds were drawn somewhat out of thin air for some of the parameters. It would have been useful if the authors had shown even a couple example plots of some of the prefilters and how thresholds were determined.

There were 25 figures in this paper, and in my opinion, many more than are useful, especially some of the earlier plots. I suggest the authors try to remove some panels in some plots, or some plots altogether, to show *representative* plots. For instance, all the noise model coefficients are given in Tables 6 & 7. Therefore, the authors can reduce Figs 3-6 to probably a single 2 or 4 panel plot (e.g., Fit Windows 2 & 3 for both GOSAT and GOSAT-2, P-polarization only). The same goes for Figs 9-12 (a single one would do) and Figs 13-16 (again, a single one would do, and not all bands are necessary). Plots are in the paper to explain findings, not to exhaustively present ever detail of the study, especially if some plots or features of plots are never discussed in the main body of the paper.

Finally, it appeared that many important previous works by other authors are never referenced, or included in the reference section but never cited in the main body. In general, referencing needs to be much improved in this work. Therefore, I recommend publication of this manuscript after a major revision to fix the issues with the burdensome # of plots and problems with referencing, as well as addressing all the specific concerns raised below.

**Specific Comments**

Section 2.3: This is a unique approach to a truth database to my knowledge – it needs more information (plots, etc) on how big this contiguous regions are / how much the TCCON data are expanded through this approach. A map of a month or a season of data density would fulfill

this, and I think be very interesting for readers. Otherwise, it's not clear how much this really expands over just using TCCON directly.

Section 2.3: Secondly, you say the requirement for contiguous regions, but you never say how close the ak-corrected CT value at the TCCON location & time has to agree with TCCON itself. Is that also 0.75 ppm? You imply this but never say – please correct this.

Section 3.1: Your terms "cloud albedo" and "water vapor path" are neither. These terms already have definitions in use by the community, and they are not how you define them. I suggest you rename "cloud albedo" to "effective albedo" or "effective scene albedo". Note you will screen out some bright desert scenes with your albedo filter, though probably not many. It looks like your 1.98  $\mu$ m filter is doing most of the work. Regarding "water vapour path", it's nothing of the sort. It's more like an SNRwv (wv=" water vapour"), or SNR1.93 (since this band is roughly at 1.93  $\mu$ m). Low SNRwv= clear, high SNRwv = cirrus present. So please rename it to something else.

Section 3.2 – Please MOTIVATE why you use both polarisations separately. Do you believe you obtain more information than if you averaged them together, or do you believe you cannot accurately average because certain instrument properties (such as ILS) are different for the two polarisations, and they themselves cannot be averaged together?

Section 3.2.1 Near line 263, you talk about the "NIR", but early in the paper you refer to ALL the bands you use as "SWIR". I realize most scientists label the O2A band as NIR and everything past 1 micron as SWIR. Can you please go through the paper and ensure consistency between NIR and SWIR labels throughout?

Section 3.2.1 – Way too many plots, as I said in the general comments. As a rule of thumb, try not to overwhelm readers with a bunch of plots that all look essentially the same. Each panel of each plot should contribute to the story you are telling.

Section 3.3.1 – In general, your "basic filter" through the RSR filters (I'm looking at your figures 1-2 for this information) really does seem basic for GOSAT, as it filters out only 8 percent of the data (35.0%-->27.2%), and most of that comes from convergence. However, for GOSAT-2 not only do twice as many soundings fail to converge as for GOSAT, but the window 5 RSR also accounts for many failed soundings (5% for GOSAT-2, versus 0.3% for GOSAT, if I am counting right). Can you please comment on why this may be happening for GOSAT-2? Window 5 is the methane band I think. You may wish to split things out separately as land versus ocean – you may find very different behaviors for the two categories. In any event, please devote a few words in this section as to why this is happening. And please do say how differently the filters act on land vs. ocean.

Actually, looking at this further, I think it is the "broadband oscillation" in the fit residuals you mention for GOSAT-2 that may be causing the problem. Are those oscillations really correlated

with retrieved XCO2 quality? If not, you may wish to loosen that constraint for GOSAT-2, to save more soundings.

Secton 3.3.2 Near line 295, please also reference Mandrake et al (2013, AMT, "Semiautonomous sounding selection for OCO-2), who did something similar for OCO-2.

Section 3.3.2, near line 310. I'm nearly certain that for water, SAA, VAA, SZA, VZA will be correlated with latitude. Because the orbit is sun-synchronous and you're looking to the glint spot over water, I'm willing to bet that any machine learning algorithm or even a simple correlation analysis can probably figure out where you are based on those quantities (or even only one or two of them). I suggest you be exceedingly careful in including those quantities. Please include a comment to this affect in the paper.

Section 3.3.2 – can you state how many training soundings total there were for GOSAT and GOSAT-2, for each of land and water? I wonder if your training set is general enough to avoid over-fitting. Also, please define "Relevance" as you use it in Figure 7 & 8.

Section 3.3.3 – This community did XCO2 bias correction long before OCO-2. Can you please reference earlier works on the subject? (The earliest I know of is Wunch et al., 2011, ACP "A method for evaluating bias..."; I believe there are similar references for GOSAT for the UoL retrieval, the NIES retrieval, and the RemoTeC retrieval).

Are you really using 10 parameters in your bias correction? This is way more than most groups usually use (which is typically 1-4; as I remember, Reuter et al. (2017) didn't use any in their OCO-2/FOCAL paper). Be careful – there could almost certainly be overfitting here. So my comment is 10 parameters simply doesn't seem to be justified based on past experience and the published methods of nearly all other retrievals for the last 10 years. Therefore, your using 10 parameters requires more justification than simply "this is what came out of the random forest algorithm".

Section 4, nearly line 378. Just a comment. The higher XCO2 variability over land has long been seen. I highly doubt this is due solely to surface variability. I think it is also caused by different scattering pathways that are not present over water. In particular, photons scattered downward by the atmosphere can be reflected off the surface back into the beam accepted by the sensor; this mechanism doesn't happen over water, so there are more ways for atmospheric scattering to degrade a retrieval. But that's mainly just a hypothesis.

Section 5 – page 13. Please include appropriate references for each of these algorithms here. Also you say for the validation of the GOSAT and GOSAT-2 FOCAL products, but really these comparisons to the other products are just for GOSAT only. You may wish to state upfront here that the vast majority of the presented validation is only for GOSAT. Only subsection 5.3 mentions GOSAT-2, and it only appears in a single validation figure (25). In fact, this paper is really begging for some basic comparison plots of GOSAT and GOSAT-2 to TCCON, to see how well your algorithm works on GOSAT-2 as compared to GOSAT. Can you please add something to that effect?

Line 438 – I do not understand this statement about a "bias anomaly". Please be more clear about what you did here. Did you subtract some kind of mean bias with respect to TCCON from each algorithm? Please don't! Or if you did, you have to state somewhere what number you subtracted off each algorithm. If ACOS is high by 1 ppm relative to TCCON and you simply subtracted that off before making plots, it's critical to state that somewhere. It would be much better simply to NOT subtract off that bias, unless you can throughly justify why you did.

Section 5.2 end, L445 – Even if you don't have "sufficient data" for full seasonal cycle fits for all GOSAT-2 data vs. TCCON, you've got enough to make some basic plots. Please do so – the community is really interested in them. If not, there isn't a lot of point in including GOSAT-2 in this paper at all.

References: It looks like you have way more references in the Reference section of the paper, than you actually reference in the main body of the paper. A rule of papers: you MUST cite each reference in your references section somewhere in the main body of the paper. Please make sure this is the case.

**Technical/Grammatical Comments**

L47: Tansat, GOSAT, and OCO-2/3 instruments  $\rightarrow$  The Tansat, GOSAT, and OCO-2/3 instruments

L280: XCO2 error is ambiguous. Suggest you change this to "XCO2 posterior uncertainty" or something more clear that it is the posterior error estimate from the OE itself, and not some error as compared to TCCON or something.

L292: Remove the word "exemplary". This isn't really an example, I assume this is a full indication of what is happening.

---

## Author Comment (AC1) · 9 Mar 2021

**Reply to referee 2**

We thank the referee for the detailed review. The comments will be considered in the revised version of the paper. In the following, the original reviewer comments are given in *italics*, our answer in normal font and the proposed updated text for the revised version of the manuscript in **bold** font.

[Figure]

Answers to major comments:

1. *Section 2.3 "True database"*

   (a) *It remains unclear whether the actual TCCON measurements are used or not in the procedure. My current understanding is that the "valid" simulations are those where the total column, computed with an homogeneous weighting and the same with the TCCON averaging kernel, differ by less than 0.75 ppm. If my understanding is correct, then (i) I do not understand why this is a valid criterium to select the most accurate simulation data and (ii) there is no need to compute the "daily mean" TCCON. If my understanding is incorrect, the description needs some re-writing.*

   For the "true" database we first determine a set of daily TCCON data for each station. These TCCON data are an average over all measurements within 13 h $\pm$ 2 h local time of one day at one station. This is no daily mean as it is only a 4 h average (we'll clarify this in the paper). From this, we get one value per day and station. We then select collocated Carbon-Tracker (CT) data for each day and station, apply the TCCON averaging kernels to them and compute XCO2. This ak-corrected CT XCO2 is then compared with the corresponding TCCON XCO2. For all days where both values agree within $\pm$ 0.75 ppm, we define contiguous regions around each station where the CT XCO2 data deviate less than 0.75 ppm from the CT value at the station. The CT data inside these regions are then inserted into the database.

   The "true" database therefore contains only CT data, which are confirmed by TCCON measurements but may differ by up to 1.5 ppm from the TCCON value. This is explicitly stated in the manuscript at the end of this section:

   "Please note that the "true" database does not contain any TCCON data - it only contains CT data which were confirmed by TCCON, but individual

values may differ by up to 1.5 ppm."

The choice of the 0.75 ppm ranges is based on a trade-off between accuracy (agreement with TCCON) and spatial coverage of the database.

We will update the section about the "true" database for clarification.

(b) *Line 138-139 "which were confirmed by TCCON" is really unclear, especially since it is said above that there is "more data in the Southern hemisphere" that is notably poor in TCCON coverage. How can such data be confirmed by TCCON.*

As explained above, "confirmed by TCCON" means that all CarbonTracker data used in the "true" reference database agree within 1.5 ppm with TC-CON. This essentially defines an area around each used TCCON station where CarbonTracker data are compliant with this criterion. The spatial coverage of the "true" database varies from day to day depending on how many data agree with TCCON. This of course also limits the spatial coverage of the "true" database (e.g. there are less data in the southern hemisphere), but nevertheless all data in the "true" database fulfil the criterion and are therefore confirmed by TCCON.

The sentence "There are typically more data in the southern hemisphere during the second half of the year" just means that there are more data during the second half of the year compared to the first half of the year, not more data in the south than in the north. We will clarify this in the text and also show example maps from the "true" data base to illustrate this.

(c) *As a consequence, it is really unclear why the database build as described, can be considered as a truth.*

For the XCO2 bias correction we have to assume some kind of "truth", and we think our choice to use a subset of CarbonTracker data which is confirmed by measurements is a valid one. We hope that with the explanations given above this is now clearer.

We also decided to rename the "true database" to "reference database" to indicate that the content is not necessarily the (essentially unknown) "true" XCO2 at a certain time and place but only an estimate which should on average reproduce large scale features correctly.

2. *Section 3.1.2 Cloud Filter*

   *It is said that the Cloud Filter is based on the fact that clouds are bright (OK) and higher in the atmosphere so that there is little water vapour above them. Then, one my expect that, when the water vapor estimate is low, a cloud presence is suspected. Yet the description of the test indicates that a cloud is detected when the water vapor estimates is \*\*larger\*\* than a threshold. This is inconsistent.*

   The referee is completely right. The term "water vapour path" which we use to name one of the filter criteria for cloudiness is misleading. As written in the manuscript, this filter is defined as the ratio between the median radiance and the median of the estimated noise, i.e. it is indeed high in case of low water vapour content and vice versa.

   We will rename "water vapour path" to "water vapour filter" to clarify this.

3. *Section 3.3.2 Random Forest Filter*

   (a) *The difference between the estimate and the "true" reference database is used. It is said that the difference is subtracted by the global monthly mean bias. This assumes that the global mean bias of the reference is zero which is a strong assumption.*

   As written in the manuscript, for the filtering we are not interested in a potential global bias of the data. The main purpose of the random forest filter is to reduce the scatter in the XCO2 data. This is why we subtract the global median of the bias before filtering. It is indeed the global median for land and water, not the monthly global median, which is subtracted – this will be

corrected in the paper. This essentially makes the filtering independent from a global bias.

For the bias correction, however, we assume that our reference (the "true" database) is on average the "truth", i.e. large scale features are well reproduced. This is indeed a strong assumption, but this is how a bias correction works, and we think we made a reasonable choice for the reference.

(b) *This section lacks a quantitative discussion: What are the mean values of the differences to the "true" reference dataset. What is the order of magnitude of the bias correction? Does it has some spatial patterns. In the case of ACOS, the bias correction is similar to the signal which is an important information. Is it the same here?*

Note that this section is not about the bias correction, it is about filtering of data. The bias correction is addressed in the following section, which also gives some quantitative information. Finally, we only use data where the estimated bias is within $\pm$ 2 ppm around the global mean bias, which is different for land and water (see Table 9). The scatter (global standard deviation) of the estimated bias is below 1 ppm.

We will add this information in the paper and also maps of the bias correction for GOSAT and GOSAT-2 such that spatial patterns can be identified.

(c) *It is then said (line 321) that the random forest classification is accurate in about two thirds of the cases. How is this evaluated? How can one decide whether it is accurate or not?*

The accuracy is estimated from the performance of the filter for the training and the test data sets (for which we know the truth). It is defined as the fraction of correctly classified samples.

We will explain this in the updated paper.

4. *Figures*

(a) *There is certainly no need for Figure 3 to 6. A couple of examples would be sufficient rather than the 28 pannels that indicate similar behaviors (and differences that are not commented).*

We agree and will replace Figs. 3 to 6 by a single one showing only an example of the noise plots for GOSAT and GOSAT-2 (two fit windows each).

(b) *Figure 9 to 12 provide no usefull information. Figure 13 to 16 could be limited to a few examples rather than the 30 pannels. I strongly recomment to combine figure 9 and 13 so that one can identify whether the fine scale structures of the residuals correspond to absorption lines*

We agree and will replace these figures by two figures (one for GOSAT and one for GOSAT-2) which will show both spectra and residuals in one plot for only one polarisation direction as indeed results for S and P are very similar.

5. *Section 4; Results*

   *This section contains several hypothetical statements "likely", "most likely", "which may explain"... that deserve investigations*

   We agree that some of our formulations are too cautious. We will check the text and update it accordingly.

Answers to other comments:

1. *Abstract : Line 21 "regional bias".*

   *There is no demonstration that TCCON is representative of a region, neither that the bias at the TCCON location is the same over a region.*

   With "regional bias" we refer to the "station-to-station bias" which is a measure for the variability of the bias between different stations and thus regions. For clarification, we will use the term "station-to-station bias" in the abstract.

2. *Section 2.2; line 111-112*

   *Why a factor of 5 for H2O "to reduce dependencies on the a-priori" but not the same factor for the CO2*

   The natural variability of H2O is much higher than for CO2, therefore we use a stronger a-priori constraint for CO2 than for H2O.

3. *Line 117 "very accurate". Please quantify.*

   The estimated accuracy of TCCON XCO2 measurements is 0.4 ppm (1 sigma). We will specify this in the text and add a reference.

4. *Line 124 : "daily mean".*

   *I understand the mean is over 4 hours. How can this be considered a daily men ?*

   Yes, the mean is indeed over 4 hours (13±2 h LT), we will correct this:

   **Then we determine from the TCCON data for each day mean values ($XCO_2^{TCCON}$) for 13 h $\pm$ 2 h local time.**

5. *Line 188 : "It is given by the ratio between the median radiance and the median of the estimated noise in this spectral range". Unclear.*

   This is the definition of the "water vapour path" filter, see also above. We will reformulate this:

   **This filter is defined as the ratio between the median radiance and the median of the estimated noise in this spectral range.**

6. *Line 278 : The case is rejected when the Angstrom coefficient is outside of the range [1 − 5]. This is strange. Clouds and aerosols can have Anstrom coefficients that are close to zero. Conversely, values larger than 2 have never been reported to my knowledge.*

The FOCAL forward model considers only a single Lambertian scattering layer to describe all scattering effects. All retrieved scattering parameters such as Ångström exponent can be considered "effective" parameters as they have to account for not only cloud/aerosol scattering but also Rayleigh scattering (which has an Ångström coefficient of 4). Because Rayleigh scattering is always present and we filter out cloudy scenes, we usually get higher effective Ångström coefficients than those expected from clouds or aerosols only. We will clarify this in the text.

7. *Line 335: "But with this filter applied". Which filter ?*

   The random forest filter, will be clarified:

   **For the bias correction we use as input the same data set as for the random forest filter, but with this random forest filter applied.**

8. *Line 343 : What is the order of magnitude of the bias correction?*

   This is explained in the following text; values are given in Table 9. We will also add maps of the bias correction, see our answer to the major comments.

9. *Line 349: "On the derived XCO2 bias".*

   *What bias is that? Is it before or after the correction? The paragraph indicates it is after correction, but then how can it be evaluated?*

   This is the bias estimated via the random forest classifier. We will clarify this:

   **These are filtered out by an additional filter on the XCO2 bias derived via the random forest classifier.**

---

## Author Comment (AC2) · 9 Mar 2021

**Reply to referee 1**

We thank the referee for the review and the constructive comments. They will be considered in the revised version of the paper. In the following, the original reviewer comments are given in *italics*, our answer in normal font and the proposed updated text for the revised version of the manuscript in **bold** font.

[Figure]

Answers to General comments:

1. *Overall, I found this paper useful and interesting, and will serve as an important reference. The subject matter is important, the layout of the paper is logical, the reasoning sound, and the results are generally laid out well. However, there are a number of problems that need to be addressed. While details of the retrieval, filtering, and bias correction were presented in a straightforward way, it was quite dry with little learned. Especially in the part about the random forest filter, which was used for both filtering and bias correction, but with little attempt on the part of the authors to explain the relevance of the features identified. The same goes for the prefilters, where it appeared that thresholds were drawn somewhat out of thin air for some of the parameters. It would have been useful if the authors had shown even a couple example plots of some of the prefilters and how thresholds were determined.*

Regarding random forest filtering, the relevance of the different features is determined by the random forest method and shown in Figs. 7 and 8. A physical explanation of these relevances is not possible – this is a general problem of this method. Therefore we think that plotting individual maps of parameters is not helpful here.

There is indeed no well defined procedure to define the prefilters, this is essentially based on "scientific knowledge", e.g. trying to set (in a physical sense) reasonable limits for some parameters. Therefore – as written in the paper – the choice of the XCO2 error limit or the maximum optical depths of the scattering layer is somewhat arbitrary and essentially based on trying out different values and looking at the resulting scatter of the data after filtering. It is therefore not possible to show e.g. a single example map of one of these parameters from which the limits can be derived. or justified.

We are aware of this problem, and for future product versions we will try to im-

prove the filtering method.

2. *There were 25 figures in this paper, and in my opinion, many more than are useful, especially some of the earlier plots. I suggest the authors try to remove some panels in some plots, or some plots altogether, to show representative plots. For instance, all the noise model coefficients are given in Tables 6 & 7. Therefore, the authors can reduce Figs 3-6 to probably a single 2 or 4 panel plot (e.g., Fit Windows 2 & 3 for both GOSAT and GOSAT-2, P-polarization only). The same goes for Figs 9-12 (a single one would do) and Figs 13-16 (again, a single one would do, and not all bands are necessary). Plots are in the paper to explain findings, not to exhaustively present ever detail of the study, especially if some plots or features of plots are never discussed in the main body of the paper.*

We agree and will combine / remove plots to reduce the number of plots accordingly. For the noise model we will show windows 2P (O2 band) and 6P (strong CO2 band) for both GOSAT and GOSAT-2 in one figure. Spectra and residuals will be combined into one plot, and we will only show P polarisation, so there will be one plot for GOSAT and one for GOSAT-2.

3. *Finally, it appeared that many important previous works by other authors are never referenced, or included in the reference section but never cited in the main body. In general, referencing needs to be much improved in this work.*

We will include additional references as suggested by the referee and also check existing references.

4. *Therefore, I recommend publication of this manuscript after a major revision to fix the issues with the burdensome # of plots and problems with referencing, as well as addressing all the specific concerns raised below.*

See below for our answers to the specific comments.

Answers to Specific comments:

1. *Section 2.3: This is a unique approach to a truth database to my knowledge – it needs more information (plots, etc) on how big this contiguous regions are / how much the TCCON data are expanded through this approach. A map of a month or a season of data density would fulfill this, and I think be very interesting for readers. Otherwise, it's not clear how much this really expands over just using TCCON directly.*

   We will add an example map of the true database.

2. *Section 2.3: Secondly, you say the requirement for contiguous regions, but you never say how close the ak-corrected CT value at the TCCON location & time has to agree with TCCON itself. Is that also 0.75 ppm? You imply this but never say – please correct this.*

   The contents of the "true" database is not selected based on a 0.75 ppm maximum difference to TCCON data but to a subset of CT data at TCCON locations. This subset has been selected as having a maximum difference of 0.75 ppm between the ak-corrected CT value at the location of the station and the TCCON value.

   The "true" database therefore contains only CarbonTracker data, which are confirmed by TCCON measurements but may differ by up to 1.5 ppm from the TCCON value. This is explicitly stated in the manuscript at the end of this section:

   "Please note that the "true" database does not contain any TCCON data - it only contains CT data which were confirmed by TCCON, but individual values may differ by up to 1.5 ppm."

   We will clarify this in the text.

   We also decided to rename the "true database" to "reference database" to indicate that the content is not necessarily the (essentially unknown) "true" XCO2 at
a certain time and place but only an estimate which should on average reproduce large scale features correctly.

3. *Section 3.1: Your terms "cloud albedo" and "water vapor path" are neither. These terms already have definitions in use by the community, and they are not how you define them. I suggest you rename "cloud albedo" to "effective albedo" or "effective scene albedo". Note you will screen out some bright desert scenes with your albedo filter, though probably not many. It looks like your 1.98 $\mu$m filter is doing most of the work. Regarding "water vapour path", it's nothing of the sort. It's more like an SNRwv(wv="water vapour"), or SNR 1.93 (since this band is roughly at 1.93 $\mu$m). Low SNRwv= clear, high SNRwv= cirrus present. So please rename it to something else.*

We agree that the nomenclature we used here is misleading. We will therefore replace "cloud albedo" by "effective albedo" and will rename "water vapour path" to "water vapour filter" to clarify this. The albedo filter may indeed remove scenes with very high surface albedo, but this seems to be uncritical in our case as we still have data over deserts after cloud filtering.

4. *Section 3.2 – Please MOTIVATE why you use both polarisations separately. Do you believe you obtain more information than if you averaged them together, or do you believe you cannot accurately average because certain instrument properties (such as ILS) are different for the two polarisations, and they themselves cannot be averaged together?*

We use both polarisation corrections mainly for the following reasons:

- In principle, information is lost when averaging S and P spectra.
- In general, the sensitivity of the instruments and therefore the calibration of the measured spectra is different for S and P. For example, as mentioned by the referee, the measured ILS is given independently for S and P.

[Figure]

- S and P include different information on scattering, which can also be used for filtering and/or bias correction.

We will mention this in the paper.

5. *Section 3.2.1 Near line 263, you talk about the "NIR", but early in the paper you refer to ALL the bands you use as "SWIR". I realize most scientists label the O2A band as NIR and everything past 1 micron as SWIR. Can you please go through the paper and ensure consistency between NIR and SWIR labels throughout?*

We will harmonise the use of "NIR" and "SWIR" in the revised version of the paper.

6. *Section 3.2.1 – Way too many plots, as I said in the general comments. As a rule of thumb, try not to overwhelm readers with a bunch of plots that all look essentially the same. Each panel of each plot should contribute to the story you are telling.*

We agree and will remove / combine figures as suggested.

7. *Section 3.3.1 – In general, your "basic filter" through the RSR filters (I'm looking at your figures 1-2 for this information) really does seem basic for GOSAT, as it filters out only 8 percent of the data (35.0%–>27.2%), and most of that comes from convergence. However, for GOSAT-2 not only do twice as many soundings fail to converge as for GOSAT, but the window 5 RSR also accounts for many failed soundings (5% for GOSAT-2, versus 0.3% for GOSAT, if I am counting right). Can you please comment on why this may be happening for GOSAT-2? Window 5 is the methane band I think. You may wish to split things out separately as land versus ocean – you may find very different behaviors for the two categories. In any event, please devote a few words in this section as to why this is happening. And please do say how differently the filters act on land vs. ocean. Actually, looking at this further, I think it is the "broadband oscillation" in the fit residuals*

*you mention for GOSAT-2 that may be causing the problem. Are those oscillations really correlated with retrieved XCO2 quality? If not, you may wish to loosen that constraint for GOSAT-2, to save more soundings.*

We use the term "basic filter" for all post-processing filters applied before the random forest filter. Indeed, the number of data removed by the RSR filter is much larger for GOSAT-2 than for GOSAT. We assume that this is mainly related to the GOSAT-2 calibration which e.g. currently does not consider remaining polarisation sensitivities to the U (45°) component of incoming light, which especially affects band 2 and may be the reasons for the broadband features in window 5.

Furthermore, as explained in the paper, there seem to be some problems with the noise of GOSAT-2 data over water, which is why we base our GOSAT-2 noise model on land data only. This is also why we cannot provide separate RSR filters for land and ocean for GOSAT-2. For GOSAT different RSR filters are not necessary as there is no discrepancy in the land/ocean noise data. This is why we apply the same convergence / RSR filters for land and water (but later split the filtering and bias correction).

We rely on the calibrated spectra given in the data product and thus cannot quantify the real impact of the deficiencies in the calibration on the retrieval results. We therefore prefer not to add speculations about possible reasons to the paper.

However, the different performance of the filters indicates that the filtering for GOSAT-2 needs some further optimisation, which is planned for the next version of the products.

We will mention the latter in the paper.

8. *Secton 3.3.2 Near line 295, please also reference Mandrake et al (2013, AMT, "Semi-autonomous sounding selection for OCO-2"), who did something similar for OCO-2.*

We will add this reference.

9. *Section 3.3.2, near line 310. I'm nearly certain that for water, SAA, VAA, SZA, VZA will be correlated with latitude. Because the orbit is sun-synchronous and you're looking to the glint spot over water, I'm willing to bet that any machine learning algorithm or even a simple correlation analysis can probably figure out where you are based on those quantities (or even only one or two of them). I suggest you be exceedingly careful in including those quantities. Please include a comment to this affect in the paper.*

We agree that for water / glint there are specific relations between viewing geometry and geolocation and will mention this in the paper.

10. *Section 3.3.2 – can you state how many training soundings total there were for GOSAT and GOSAT-2, for each of land and water? I wonder if your training set is general enough to avoid over-fitting. Also, please define "Relevance" as you use it in Figure 7 & 8.*

The number of soundings used for the random forest filter are:

- GOSAT land: 54317
- GOSAT water: 109414
- GOSAT-2 land: 10625
- GOSAT-2 water: 40459

From these, we use 90% for training and 10% for testing, so this should be sufficient for 10 parameters. We think that over-fitting is not a problem here, because as mentioned in the paper we only get accurate filter results in two thirds of the cases (for both training and test data sets). This would not be the case for over-fitting where the training set would have a much better performance than the test data set.

"Relevance" is a quantity coming out of the random forest method which describes the relative importance of each feature for the filtering. Relevances are normalised such that the sum of all relevances is 1.

We will include this information in the paper.

11. *Section 3.3.3 – This community did XCO2 bias correction long before OCO-2. Can you please reference earlier works on the subject? (The earliest I know of is Wunch et al., 2011, ACP "A method for evaluating bias..."; I believe there are similar references for GOSAT for the UoL retrieval, the NIES retrieval, and the RemoTeC retrieval). Are you really using 10 parameters in your bias correction? This is way more than most groups usually use (which is typically 1-4; as I remember, Reuter et al.(2017) didn't use any in their OCO-2/FOCAL paper). Be careful – there could almost certainly be overfitting here. So my comment is 10 parameters simply doesn't seem to be justified based on past experience and the published methods of nearly all other retrievals for the last 10 years. Therefore, your using 10 parameters requires more justification than simply "this is what came out of the random forest algorithm".*

We will add additional references to other bias correction methods.

The choice of filter parameters is mainly based on the output of the random forest method, which associates to all possible parameters a relevance value (shown e.g. in the top of Fig. 7 for GOSAT). As explained above, this relevance value describes the relative importance of a suggested feature for the filtering or regression. As can be seen from this figure, the relevance drops off quite rapidly after a few variables. The number of variables to be used is then a trade-off between many variables (explaining all relations with a risk for over-fitting and high computational effort) and few variables (no over-fitting, but maybe missing some relations). We decided to use 10 variables as a good compromise, and since we do not seem to have problems with over-fitting (see answer to previous comment) this seems justified. Note that even if we would include too many non-relevant parameters, this would be no major problem as most filtering is done by the relevant parameters.

We will include this explanation in the paper.

12. *Section 4, nearly line 378. Just a comment. The higher XCO2 variability over land has long been seen. I highly doubt this is due solely to surface variability. I think it is also caused by different scattering pathways that are not present over water. In particular, photons scattered downward by the atmosphere can be reflected off the surface back into the beam accepted by the sensor; this mechanism doesn't happen over water, so there are more ways for atmospheric scattering to degrade a retrieval. But that's mainly just a hypothesis.*

Thank you for the information. We agree that surface variability may not be the only reason for the higher XCO2 variability over land. We will adapt the text accordingly and take your comment into account.

13. *Section 5 – page 13. Please include appropriate references for each of these algorithms here. Also you say for the validation of the GOSAT and GOSAT-2 FOCAL products, but really these comparisons to the other products are just for GOSAT only. You may wish to state upfront here that the vast majority of the presented validation is only for GOSAT. Only subsection 5.3 mentions GOSAT-2, and it only appears in a single validation figure (25). In fact, this paper is really begging for some basic comparison plots of GOSAT and GOSAT-2 to TCCON, to see how well your algorithm works on GOSAT-2 as compared to GOSAT. Can you please add something to that effect?*

Detailed references for all products are already given in Section 2.4. Indeed, most comparisons are done for GOSAT, except for the time series (section 5.3).

We did not include a comparison of GOSAT-2 with TCCON because of the limited amount of the GOSAT-2 data. We will add this in the revised version but note also that the results are probably not representative.

14. *Line 438 – I do not understand this statement about a "bias anomaly". Please be more clear about what you did here. Did you subtract some kind of mean bias with respect to TCCON from each algorithm? Please don't! Or if you did, you have to state somewhere what number you subtracted off each algorithm. If ACOS is high by 1 ppm relative to TCCON and you simply subtracted that off before making plots, it's critical to state that somewhere. It would be much better simply to NOT subtract off that bias, unless you can throughly justify why you did.*

We subtracted for each algorithm the mean of the bias for all stations. This is because for most applications this mean bias is not relevant since most information is contained in gradients. Subtracting a mean bias also facilitates a comparison of different bias patterns between the algorithms.

The subtracted mean station bias is actually small (0.17 – 0.64 ppm). We will mention this in the paper.

15. *Section 5.2 end, L445 – Even if you don't have "sufficient data" for full seasonal cycle fits for all GOSAT-2 data vs. TCCON, you've got enough to make some basic plots. Please do so – the community is really interested in them. If not, there isn't a lot of point in including GOSAT-2 in this paper at all.*

As mentioned above, we will include some plots for GOSAT-2 vs. TCCON.

16. *References: It looks like you have way more references in the Reference section of the paper, than you actually reference in the main body of the paper. A rule of papers: you MUST cite each reference in your references section somewhere in the main body of the paper. Please make sure this is the case.*

We will check the references, but actually all entries in the References section are cited in the paper. However, the references for the used TCCON data are only listed in Table 1, maybe this is why the referee misses them in the main text.

Answers to Technical/Grammatical comments:

1. *47: Tansat, GOSAT, and OCO-2/3 instruments The Tansat, GOSAT, and OCO-2/3 instruments*

   Will be corrected.

2. *L280: XCO2 error is ambiguous. Suggest you change this to "XCO2 posterior uncertainty" or something more clear that it is the posterior error estimate from the OE itself, and not some error as compared to TCCON or something.*

   Agreed, will be changed.

3. *L292: Remove the word "exemplary". This isn't really an example, I assume this is a full indication of what is happening.*

   Indeed, "exemplary" may not be the correct term here. However, the figures only show an example for one month of data (as written in the caption). We will rephrase this sentence:

   **Figs. 1 and 2 show how many data points are typically filtered out in this step.**

   Note: The numbering of figures above refers to the current paper, it will change in the revised version.
* * *